# Ubiquitination of phytoene synthase 1 precursor modulates carotenoid biosynthesis in tomato

Peiwen Wang[1,2,3], Yuying Wang[1,3], Weihao Wang[1], Tong Chen[1], Shiping Tian[1,2] & Guozheng Qin [1,2✉]

Carotenoids are natural pigments that are indispensable to plants and humans, whereas the regulation of carotenoid biosynthesis by post-translational modification remains elusive. Here, we show that a tomato E3 ubiquitin ligase, Plastid Protein Sensing RING E3 ligase 1 (PPSR1), is responsible for the regulation of carotenoid biosynthesis. PPSR1 exhibits self-ubiquitination activity and loss of PPSR1 function leads to an increase in carotenoids in tomato fruit. PPSR1 affects the abundance of 288 proteins, including phytoene synthase 1 (PSY1), the key rate-limiting enzyme in the carotenoid biosynthetic pathway. PSY1 contains two ubiquitinated lysine residues (Lys380 and Lys406) as revealed by the global analysis and characterization of protein ubiquitination. We provide evidence that PPSR1 interacts with PSY1 precursor protein and mediates its degradation via ubiquitination, thereby affecting the steady-state level of PSY1 protein. Our findings not only uncover a regulatory mechanism for controlling carotenoid biosynthesis, but also provide a strategy for developing carotenoid-enriched horticultural crops.

[1] Key Laboratory of Plant Resources, Institute of Botany, the Innovative Academy of Seed Design, Chinese Academy of Sciences, No. 20 Nanxincun, Xiangshan, Haidian District, Beijing 100093, China. [2] University of Chinese Academy of Sciences, Beijing 100049, China. [3] These authors contributed equally: Peiwen Wang, Yuying Wang. ✉email: gzqin@ibcas.ac.cn

Carotenoids are a group of 40-carbon isoprenoid compounds that are synthesized by all photosynthetic organisms (bacteria, algae, and plants) and some non-photosynthetic bacteria and fungi[1,2]. As natural pigments, carotenoids are indispensable to plants and humans. In plants, carotenoids act as structural components of photosynthetic machinery, protect plants from photooxidative damage, and serve as precursors for phytohormones (abscisic acid and strigolactones) and other signaling molecules[1,3,4]. Carotenoids are also important contributors to fruit color and nutritional quality in horticultural crops such as tomato and citrus, which accumulate carotenoids during fruit ripening[5–7]. In humans, carotenoids provide precursors for biosynthesis of vitamin A and serve as antioxidants[4]. Epidemiological analyses demonstrate that dietary intake of carotenoid-rich foods can lower the risk of degenerative diseases[1,8].

Due to the pivotal role of carotenoids in nature, molecular dissection of carotenoid metabolism and its regulatory network has received considerable interest. The biosynthesis of carotenoids is a complicated process that involves a series of steps[9]. The genes responsible for carotenoid biosynthesis in higher plants have been well-defined[1]; however, the regulatory mechanisms that govern their action, especially the regulation at post-translational level, are poorly understood.

Plant carotenoids are synthesized in plastids, a type of plant-specific organelles, of which the typical members include chloroplasts and chromoplasts[1,10]. Plastids are semi-autonomous organelles and more than 90% of the proteins in the plastid, including those in the carotenoid biosynthetic pathway, are nucleus-encoded[10]. These plastid-targeted proteins are synthesized in the cytosol as preproteins (precursor proteins), which contain an N-terminal transit peptide that directs the precursor proteins into the plastid before being proteolytically removed[11,12]. Recently, it was shown that protein ubiquitination, an essential post-translational modification, mediates plastid precursor protein degradation[12,13], but how this affects relevant physiological process in the plastid remain uncertain. Moreover, whether precursors of proteins in specific metabolic processes such as the carotenoid biosynthetic pathway are regulated by ubiquitination has not been defined.

Protein ubiquitination involves the concerted action of a cascade of enzymes consisting of the ubiquitin-activating enzyme (E1), ubiquitin-conjugating enzyme (E2), and ubiquitin ligase (E3)[14]. Substrate specificity is determined primarily by the E3 ubiquitin ligases, which enable specific recognition and regulation of numerous, functionally diverse substrates[15]. In the present study, we identify a tomato E3 ubiquitin ligase, Plastid Protein Sensing RING E3 ligase 1 (PPSR1), in a screening of the interacting proteins of SlUBC32, an E2 enzyme that has been shown in our previous work to be involved in fruit ripening[16]. Mutation of *PPSR1* by CRISPR/Cas9 gene-editing system affects carotenoid biosynthesis in tomato fruit. We demonstrate that PPSR1 recognizes the precursor of phytoene synthase 1 (PSY1), the main rate-limiting enzyme in the carotenoid biosynthetic pathway, and mediates its ubiquitination and degradation. Our study reveals a direct role for protein ubiquitination in the regulation of carotenoid biosynthesis.

## Results

### PPSR1 directly interacts with SlUBC32.
To get insight into the molecular basis of SlUBC32 in regulating fruit ripening, we performed a yeast two-hybrid (Y2H) screen to identify proteins that interact with SlUBC32 using a tomato cDNA library. A putative really interesting new gene (RING) E3 ubiquitin ligase (Solyc01g006810), which we named PPSR1, was identified as the candidate SlUBC32-interacting protein. Y2H analysis confirmed the interactions between PPSR1 and SlUBC32 (Fig. 1a). We then carried out split luciferase complementation imaging (LCI) assay, in which cLUC-PPSR1 and SlUBC32-nLUC were transiently co-expressed in leaves of tobacco (*Nicotiana benthamiana*). An intense luciferase activity was detected in tobacco leaves co-expressing cLUC-PPSR1 and SlUBC32-nLUC, whereas the negative controls showed no luciferase activity (Fig. 1b), indicating that PPSR1 interacts with SlUBC32. We subsequently investigated whether PPSR1 interacts with SlUBC32 in vitro. The MBP-tagged PPSR1 (MBP-PPSR1) and HA-tagged SlUBC32 (SlUBC32-HA) recombinant proteins purified from *Escherichia coli* were mixed and incubated with anti-HA agarose, and then the precipitated products were examined by immunoblot analysis. As shown in Fig. 1c, SlUBC32-HA can directly bind to MBP-PPSR1 in vitro, but not MBP tag protein. To further verify the interactions between PPSR1 and SlUBC32, a co-immunoprecipitation (Co-IP) assay was conducted in leaves of *N. benthamiana* co-expressing Flag-tagged PPSR1 (Flag-PPSR1) and HA-tagged SlUBC32 (SlUBC32-HA). It was shown that Flag-PPSR1 was immunoprecipitated with SlUBC32-HA by anti-HA agarose, whereas no signal was observed in the case of Flag-PPSR1 or SlUBC32-HA alone (Fig. 1d). Notably, high molecular weight signals over a band of the predicted Flag-PPSR1 (indicated by a red arrowhead) was observed in the input of Co-IP (Fig. 1d), which may be caused by the self-ubiquitination of PPSR1. Co-expression resulted in an increase in amounts of Flag-PPSR1 and this could be explained by the feedback regulation of PPSR1 by transcriptional regulators that are targeted by PPSR1. It is conceivable that co-expression decreases PPSR1 activity due to self-ubiquitination, thereby attenuating the degradation of the substrates including the transcriptional regulators, which in turn improve the expression of PPSR1. Taken together, these data demonstrated that PPSR1 interacts with SlUBC32.

To explore the intracellular colocalization of PPSR1 and SlUBC32, their coding sequences were introduced into a plasmid to generate a translational fusion with enhanced green fluorescent protein (eGFP) and monomeric Cherry protein (mCherry), respectively. The results showed that the fluorescent signals of eGFP-tagged PPSR1 (PPSR1-eGFP) colocalized with those of mCherry-tagged SlUBC32 (SlUBC32-mCherry) in the cytosol and nucleus (Fig. 1e), suggesting the subcellular colocalization of PPSR1 and SlUBC32.

### PPSR1 is a RING-type E3 ubiquitin ligase with self-ubiquitination activity.
As a putative RING-type E3 ubiquitin ligase, the function of PPSR1 in tomato remains uncharacterized. PPSR1 is composed of 342 amino acids, containing a RING-type zinc finger domain (RING domain) in its N-terminal region (Fig. 2a). According to alignment of homologous protein sequences, the RING domain in PPSR1 is a C3H2C3-type zinc finger whose cysteine (C) and histidine (H) residues can chelate two zinc ions[17]. *PPSR1* gene was expressed in both vegetative and reproductive organs including roots, stems, leaves, flowers, and fruits, and declined gradually during fruit ripening (Supplementary Fig. 1a). In contrast, PPSR1 protein exhibited a relatively stable state in fruit (Supplementary Fig. 1b). RING domain-contained proteins generally have E3 ubiquitin ligase activity[18,19]. To determine whether PPSR1 functions as an E3 ligase, the MBP-tagged recombinant PPSR1 protein (MBP-PPSR1) was purified from *E. coli* and subjected to in vitro ubiquitination assay by incubation with wheat E1, human E2, and *Arabidopsis* ubiquitin. The reaction products were detected by immunoblot analysis using anti-MBP and anti-ubiquitin antibodies, respectively. As shown in Fig. 2b and Supplementary Fig. 2, the signals of high

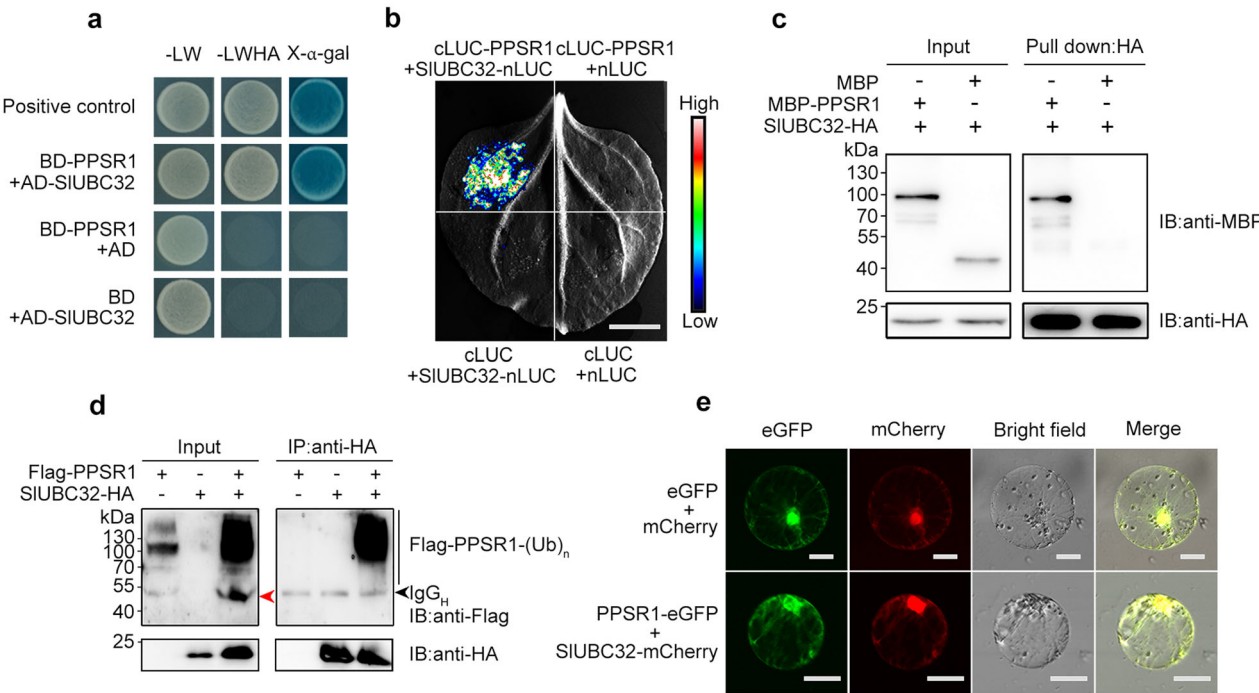

**Fig. 1 PPSR1 interacts with SlUBC32. a** Y2H assay revealing the interactions between PPSR1 and SlUBC32. The PPSR1 fused with the binding domain (BD) of GAL4 (BD-PPSR1) and the SlUBC32 fused with the activation domain (AD) of GAL4 (AD-SlUBC32) were co-expressed in yeast. The transformants were selected on SD/-Leu/-Trp (-LW) and SD/-Leu/-Trp/-His/-Ade (-LWHA) with or without X-α-gal. **b** LCI assay revealing the interactions between PPSR1 and SlUBC32. The PPSR1 fused with the C-terminus of LUC (cLUC-PPSR1) was co-expressed with the SlUBC32 fused with the N-terminus of LUC (SlUBC32-nLUC) in tobacco (*Nicotiana benthamiana*) leaves. Scale bar, 1 cm. **c** Pull-down assay revealing the interactions between PPSR1 and SlUBC32. The recombinant SlUBC32-HA, MBP-PPSR1, and MBP (negative control) were mixed as indicated, and incubated with anti-HA agarose. The eluted proteins were detected by immunoblot using anti-MBP and anti-HA antibodies, respectively. IB, immunoblot. **d** Co-IP assay revealing the interactions between PPSR1 and SlUBC32. The Flag-PPSR1 and SlUBC32-HA fusion proteins were co-expressed in *N. benthamiana* leaves. The total proteins were extracted from the infected leaves treated with MG132 and immunoprecipitated by anti-HA agarose. The eluted proteins were then detected by immunoblot using anti-Flag and anti-HA antibodies, respectively. The red arrowhead indicates the predicted Flag-PPSR1. The black arrowhead refers to heavy chain of antibody (IgG$_{\text{H}}$). (Ub)$_n$, polyubiquitin chain. **e** Subcellular colocalization of PPSR1 and SlUBC32. The *Agrobacteria* carrying 35S::*PPSR1-eGFP* and 35S::*SlUBC32-mCherry* constructs were transiently co-transformed into tobacco leaves. The tobacco protoplasts co-expressing eGFP and mCherry were used as negative control. Scale bars, 20 μm.

molecular mass bands, which represent ubiquitinated proteins, were observed in the intact reaction system using both anti-MBP and anti-ubiquitin detection, but not in the absence of a single component. These data indicated that PPSR1 has E3 ubiquitin ligase activity in vitro and can catalyze its self-ubiquitination. When the intact PPSR1 was substituted by the mutated form (mtPPSR1) in which conserved cysteine (C49) and histidine (H51 and H54) residues in the RING domain were replaced by serine (S) and tyrosine (Y), respectively (Fig. 2a), the bands for the ubiquitinated proteins failed to appear (Fig. 2c), demonstrating the critical role of the RING domain for the E3 ligase activity.

In order to test whether PPSR1 acts in cooperation with ubiquitin E2 enzyme SlUBC32, the purified recombinant SlUBC32-HA, in combination with purified E1, ubiquitin, and MBP-PPSR1, was used for in vitro ubiquitination assay. As shown in Fig. 2d, the signals of ubiquitinated MBP-PPSR1 were detected in the reaction products, but appeared to be blurry in the negative control, demonstrating that PPSR1 cooperates with SlUBC32 to accomplish protein ubiquitination. The self-ubiquitination activity may explain the stable state of PPSR1 protein level during fruit ripening. At the early stage of fruit ripening, the mRNA level of *PPSR1* is high, but part of the translated PPSR1 protein undergoes degradation via self-ubiquitination due to the absence of substrate proteins. At the later stage of fruit ripening, although the mRNA level of *PPSR1* is decreased, the accumulation of substrate

proteins alleviates the self-ubiquitination of PPSR1 protein, thus maintaining the stable state of PPSR1[20].

Previous studies have shown that RING-type E3 ubiquitin ligases usually function in the form of dimer[19,20]. To examine whether this happens in PPSR1, Y2H analysis was performed and the results revealed that PPSR1 proteins interact with each other (Fig. 2e). The self-interactions of PPSR1 proteins were verified by LCI assay, in which the luciferase activity was detected when cLUC-PPSR1 and PPSR1-nLUC were co-expressed in tobacco leaves (Fig. 2f). These data suggest the existence of dimerization for PPSR1 proteins.

Given PPSR1 exhibits self-ubiquitination activity that may sharply reduce its stability[20], we measured the half-life of PPSR1 using a cell-free degradation assay. Protein degradation was observed for MBP-PPSR1 and its half-life arrived at about 2.5 h of incubation (Fig. 2g, h). By comparison, the protein levels of MBP-mtPPSR1 remained more than 70% of the initial content at that time, indicating that the degradation rate of MBP-PPSR1 is markedly faster than that of its mutant form (Fig. 2g, h). To assess the stability of PPSR1 in vivo, Flag-PPSR1 and Flag-mtPPSR1 were transiently expressed in tobacco leaves, respectively. Immunoblot analysis revealed that the levels of Flag-mtPPSR1 were higher than those of Flag-PPSR1, and both fusion proteins were accumulated after treatment with MG132, a 26S proteasome inhibitor (Fig. 2i). Collectively, these data pointed out that the RING domain is

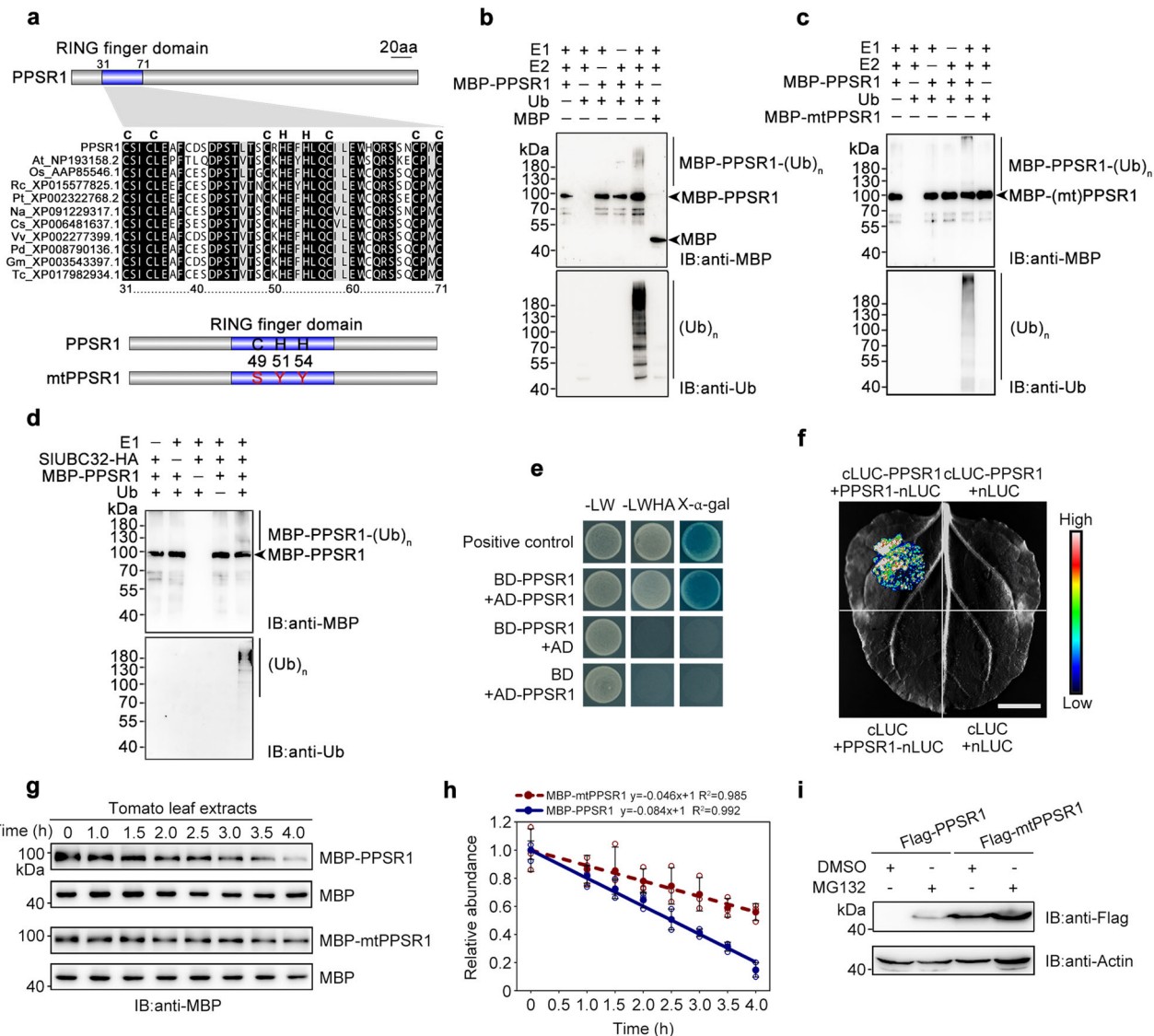

**Fig. 2 PPSR1 exhibits self-ubiquitination activity. a** Characterization of the RING finger domain in PPSR1. Sequence alignment of the RING finger domain reveals the conserved amino acids. The mutated form of PPSR1 (mtPPSR1) was generated by site-directed mutagenesis. C, cysteine; H, histidine; S, serine; Y, tyrosine. At, *Arabidopsis thaliana*; Os, *Oryza sativa*; Rc, *Ricinus communis*; Pt, *Populus trichocarpa*; Na, *Nicotiana attenuata*; Cs, *Citrus sinensis*; Vv, *Vitis vinifera*; Pd, *Phoenix dactylifera*; Gm, *Glycine max*; Tc, *Theobroma cacao*. **b, c** Ubiquitination assay of PPSR1 in vitro. An ubiquitination reaction was carried out in the presence (+) or absence (−) of E1, E2, ubiquitin (Ub), and MBP-PPSR1 (**b**) or MBP-mtPPSR1 (**c**). The reaction products were subjected to immunoblot using anti-MBP and anti-Ub antibodies, respectively. MBP protein was used as the negative control in (**b**). (Ub)$_n$, polyubiquitin chain. IB, immunoblot. **d** PPSR1 ubiquitination assay using SlUBC32 instead of E2 used in (**b**). **e** Y2H assay revealing the self-interaction of PPSR1. The PPSR1 fused with the activation domain (AD) of GAL4 (AD-PPSR1) and the binding domain (BD) of GAL4 (BD-PPSR1) were co-expressed in yeast. The transformants were selected on SD/-Leu/-Trp (-LW) and SD/-Leu/-Trp/-His/-Ade (-LWHA) with or without X-α-gal. **f** LCI assay revealing the self-interaction of PPSR1. The PPSR1 fused with the C-terminus of LUC (cLUC-PPSR1) was co-expressed with the PPSR1 fused with the N-terminus of LUC (PPSR1-nLUC) in tobacco (*Nicotiana benthamiana*) leaves. Scale bar, 1 cm. **g** Cell-free degradation assay of PPSR1. The recombinant MBP-PPSR1 and MBP-mtPPSR1 proteins were purified and incubated in the extracts from tomato leaves, respectively. The protein levels were measured by immunoblot using an anti-MBP antibody at different time intervals. The MBP protein was used as the loading control. **h** Quantification of protein levels in (**g**) by ImageJ. Error bars represent the means ± standard deviation (SD) of three independent experiments. The circles indicate individual data points. **i** Stability analysis of PPSR1. The Flag-PPSR1 and Flag-mtPPSR1 fusion proteins were expressed in *N. benthamiana* leaves, respectively. The protein levels were determined by immunoblot using an anti-Flag antibody. The *N. benthamiana* actin was used as the loading control.

required for the E3 ligase activity of PPSR1, which may function as dimers and undergo degradation via self-ubiquitination.

**Loss of PPSR1 function increases carotenoid accumulation in tomato fruit.** To investigate the function of PPSR1 in tomato, we generated *ppsr1* mutants using a CRISPR/Cas9 gene-editing

system. Four single guide RNAs (sgRNAs) that contain different target sequences (T1, T2, T3, and T4) were designed to specifically target the exons of *PPSR1* (Fig. 3a). Among transgenic plants in the second generation, three distinct homozygous mutant lines (*ppsr1-4*, *ppsr1-10*, and *ppsr1-13*) were isolated and confirmed by sequencing genomic regions flanking the target sites. These homozygous mutants carry 7-bp deletion (*ppsr1-4*), 1-bp insertion

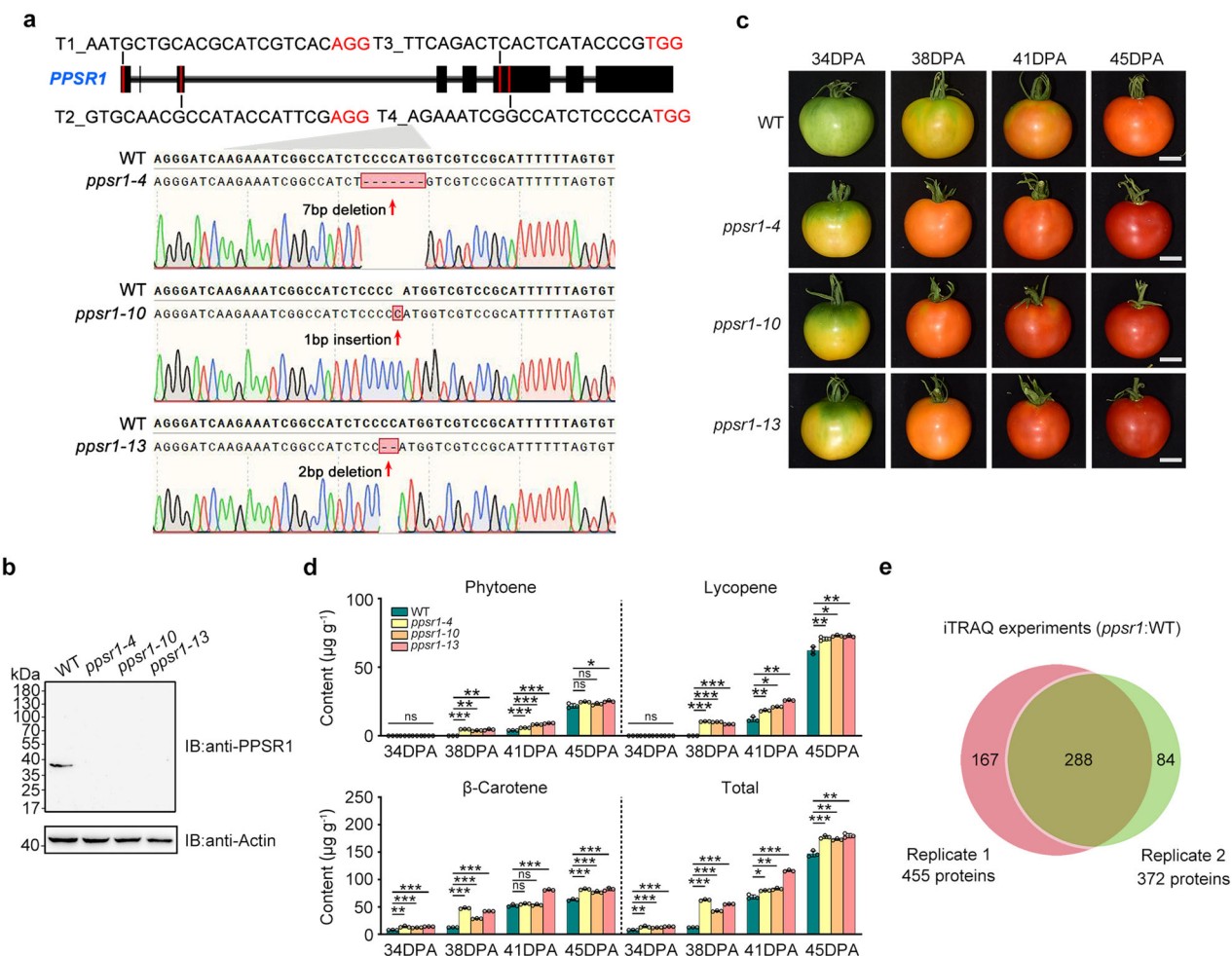

**Fig. 3 PPSR1 modulates carotenoid accumulation in tomato fruit during ripening. a** Genotyping of mutations mediated by CRISPR/Cas9 gene-editing system in *ppsr1-4*, *ppsr1-10*, and *ppsr1-13* mutants. Schematic illustration shows the single guide RNAs (sgRNAs) containing different target sequences (T1, T2, T3, and T4) that were designed to specifically target the exons of *PPSR1*. Red letters represent the protospacer adjacent motif (PAM). Red arrows indicate the editing sites that were verified by sequencing. **b** Absence of PPSR1 protein in the *ppsr1* mutants. Total proteins were extracted from fruit of wild-type (WT) and *ppsr1* mutants at 38 days post-anthesis (DPA) and subjected to immunoblot using an anti-PPSR1 antibody. Equal loading was confirmed by an anti-actin antibody. IB, immunoblot. **c** Phenotype analysis of *ppsr1* mutants. Fruit from WT and *ppsr1* mutants at 34, 38, 41, and 45 DPA are shown. Scale bars, 2 cm. **d** Accumulation of carotenoids (phytoene, lycopene, and β-carotene) in fruit of WT and *ppsr1* mutants during ripening. Error bars represent the means ± standard deviation (SD) of three independent experiments. The circles indicate individual data points. Asterisks indicate significant differences (*$P < 0.05$, **$P < 0.01$, ***$P < 0.001$; two-tailed Student's *t*-test). **e** Venn diagram showing the overlap of proteins that exhibit differential expression in the *ppsr1* mutant fruit compared to the WT in two independent biological replicates of quantitative proteome analysis. Proteins isolated from WT and *ppsr1* mutant fruit at 38 DPA were subjected to iTRAQ (isobaric tags for relative and absolute quantification) labeling coupled with NanoLC–MS/MS.

(*ppsr1-10*), or 2-bp deletion (*ppsr1-13*) caused by target T4 in the sixth exon of *PPSR1* (Fig. 3a), and no editing events occur around the sequence of target T1/2/3. All mutants were predicted to cause premature termination of PPSR1 protein translation within the following 40-bp sequence of editing sites. Immunoblot analysis detected a band corresponding to the predicted size (~38-kDa) of the full-length PPSR1 only in the wild type (Fig. 3b). No bands were observed in the *ppsr1* mutants, indicating that the predicted truncated versions of PPSR1 (~19-kDa) did not generate and *PPSR1* was successfully knocked out in three *ppsr1* mutants. The potential off-target sites in the tomato genome were predicted by CRISPR-P (version 2.0, http://crispr.hzau.edu.cn/CRISPR2/), and no mutagenesis was found in the six potential off-target sites (Supplementary Fig. 3), suggesting the specific mutation for *PPSR1*.

The *ppsr1* mutant lines (*ppsr1-4*, *ppsr1-10*, and *ppsr1-13*) showed obvious and similar ripening-accelerated phenotypes (Fig. 3c). A visible color change was observed at 34 days post-anthesis (DPA) in the *ppsr1* mutant fruit, while the wild-type

tomato remained green at this stage (Fig. 3c). At 38 DPA, the *ppsr1* mutant fruit displayed a homogenous orange color, whereas the fruit from the wild type was only just starting to change color. This suggests that *PPSR1* is responsible for tomato fruit pigmentation. Detection of the content of three important carotenoids (phytoene, lycopene, and β-carotene) in tomato fruit indicated that the accelerated coloration in *ppsr1* fruit correlated with the significantly elevated carotenoid contents (Fig. 3d).

E3 ubiquitin ligase mediates protein degradation, leading to the changes in protein abundance[21]. To identify the proteins that differentially accumulate in the *ppsr1* mutant, we performed a quantitative proteomic analysis using iTRAQ (isobaric tags for relative and absolute quantification) approach. Proteins isolated from *ppsr1* mutant fruit and wild-type fruit at 38 DPA were labeled with iTRAQ reagents and submitted to NanoLC–MS/MS analysis. A total of 5318 and 5375 proteins were identified in two independent biological replicates, respectively, with a global false discovery rate (FDR) below 1% in both. Quantitative analysis

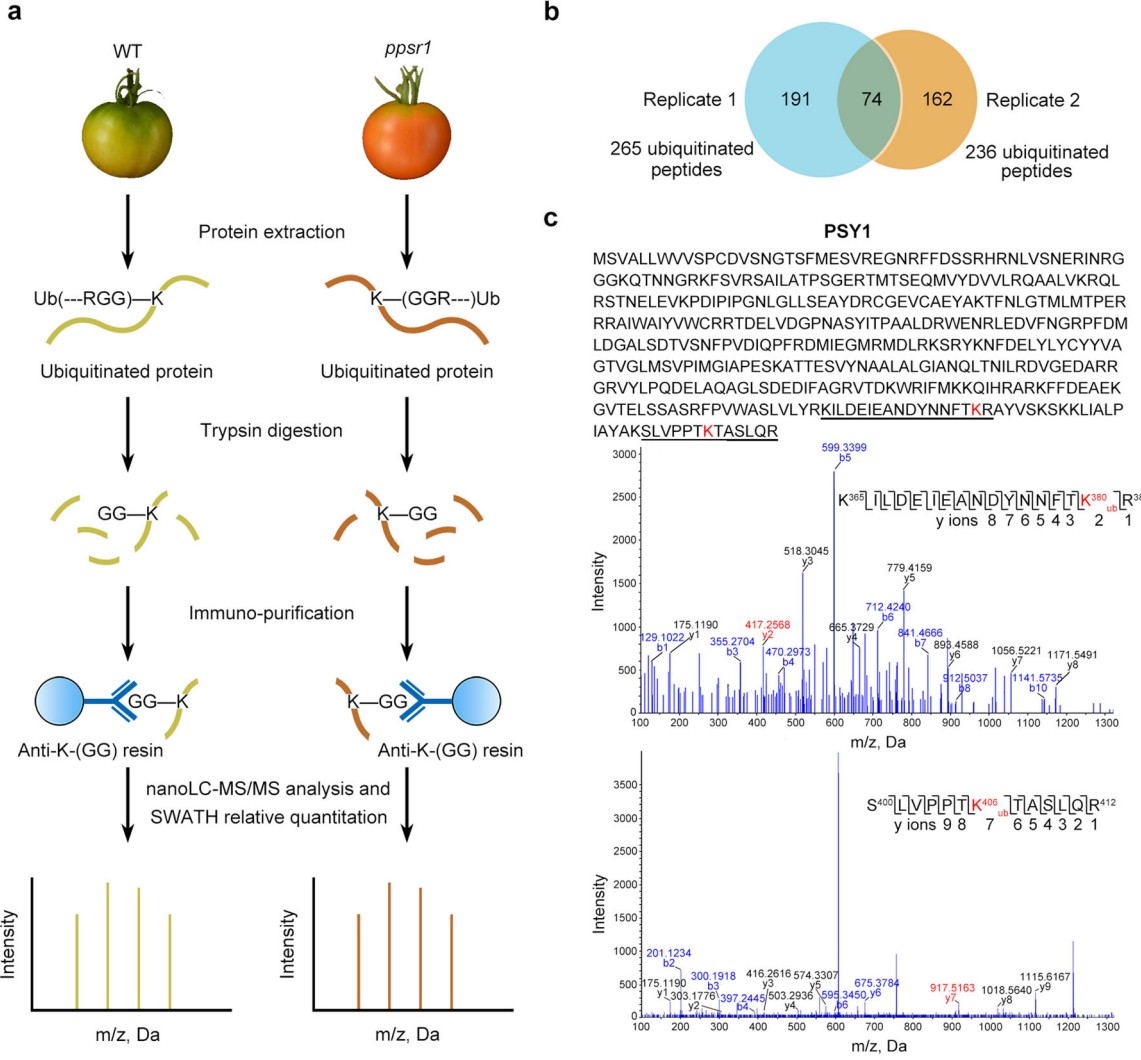

**Fig. 4 Identification of PSY1 as the candidate substrate of PPSR1. a** A workflow diagram showing the identification and quantification of ubiquitinated peptides that exhibit significant differences in abundance in the *ppsr1* mutant fruit compared to the wild type (WT). Proteins isolated from WT and *ppsr1* mutant fruit at 38 DPA were digested with trypsin, followed by immunoprecipitation with an anti-K-(GG) antibody. The recovered ubiquitinated peptides were submitted to SWATH-MS (Sequential Window Acquisition of all Theoretical Mass Spectra) quantitative proteomic analysis. **b** Overlap of the ubiquitinated peptides identified in two independent biological replicates. **c** Identification of ubiquitination sites in PSY1 by NanoLC–MS/MS. Sequences of the identified ubiquitinated peptides are underlined in the PSY1 protein sequence. The mass spectra of two ubiquitinated peptides are displayed. The y-ions and the corresponding peptide sequence are presented, with ubiquitinated lysine (K) residue marked in red.

indicated that 288 proteins were differentially expressed in the *ppsr1* mutant fruit compared to the wild type in both biological replicates (Fig. 3e). Supplementary Data 1 shows these proteins along with the relevant identification information. We identified a number of ripening-related proteins, of which several are involved in carotenoid biosynthesis, including carotenoid isomerase (CRTISO), PSY1, and phytoene desaturase (PDS), and the levels of these proteins were higher in the *ppsr1* mutant fruit (Supplementary Data 1), consistent with the carotenoid-enhanced phenotype.

**PSY1 is identified as a candidate substrate of PPSR1.** E3 ubiquitin ligases mediate the degradation of substrate proteins via ubiquitination, which involves the attachment of ubiquitin to lysine (K) residues on substrate proteins[18]. We next sought to identify PPSR1 substrates and their ubiquitination sites by using a state-of-the-art technique, which couples a K-(GG) peptide immunoprecipitation with quantitative proteomic analysis

(Fig. 4a). The proteins extracted from wild-type and *ppsr1* mutant fruit at 38 DPA were trypsin digested and immunoprecipitated with an anti-K-(GG) antibody that specifically enriches for peptides harboring lysine residues modified by diglycine (Gly-Gly, diGly), an adduct left at sites of ubiquitination after trypsin digestion[22]. The recovered diGly peptides, which represent ubiquitinated peptides, were analyzed by SWATH-MS (Sequential Window Acquisition of all Theoretical Mass Spectra), a quantitative proteomic approach[23], which has been successfully used to measure quantitative changes of *N*-linked glycoproteins[24]. The ubiquitination sites, i.e., ubiquitinated lysine residues, were identified using MS/MS based on their retention time, mass-to-charge ratio (*m/z*) and charge states. Two independent biological replicates identified 265 and 236 diGly peptides, respectively. Seventy-four diGly peptides were overlapped in both biological replicates (Fig. 4b). Quantitative analysis revealed that 27 of these diGly peptides changed abundance significantly (*P* < 0.05) in the *ppsr1* mutant (Supplementary Data 2). We focused on diGly peptides with lower abundance in the *ppsr1* mutant, because

mutation of *PPSR1* was speculated to decrease the levels of ubiquitinated peptides from substrate proteins. This resulted in the identification of 24 diGly peptides containing 24 ubiquitination sites from 18 proteins, including PSY1, E8, alcohol dehydrogenase 2 (ADH2), ClpB chaperone (CLPB), and outer membrane lipoprotein blc (BLC) (Fig. 4c; Supplementary Fig. 4). Most of these proteins exhibited equal or higher abundance in the *ppsr1* mutant fruit compared to the wild type as revealed by the iTRAQ proteome analysis (Supplementary Data 1), suggesting that they might serve as the substrates of PPSR1. Surprisingly, two ubiquitinated lysine residues (Lys380 and Lys406) were identified in PSY1 (Fig. 4c), which represents the key enzyme in the carotenoid biosynthetic pathway. PSY1 functions in the plastid, where protein degradation is supposed to be mainly mediated by protease complex such as Clp and FtsH protease complex instead of the ubiquitin–proteasome system[25,26]. Since PSY1 is synthesized in the cytosol as precursor proteins and then imported into the plastid, we hypothesize that PPSR1 mediates ubiquitination and degradation of PSY1 precursor proteins, which may accumulate in the cytosol upon plastid conversion during fruit ripening.

**PPSR1 interacts with PSY1 precursor and mediates its ubiquitination**. To test our hypothesis that PSY1 is a substrate of PPSR1, we first assessed the interactions between PPSR1 and PSY1 by Y2H analysis. As a plastid-localized protein, PSY1 contains a putative chloroplast transit peptide (cTP; 1–62 amino acids), which may interfere with the Y2H assay, in its N-terminal region according to the prediction of TargetP v1.1 (http://www.cbs.dtu.dk/services/TargetP-1.1/index.php) (Fig. 5a). It was shown that PSY1 without transit peptide (PSY1$^{63–412}$) interacted with PPSR1 (Fig. 5b). We then examined the region of PSY1 that is needed for the interactions. PSY1$^{63–412}$ was truncated into three fragments, namely PSY1$^{130–412}$, PSY1$^{239–412}$, and PSY1$^{130–238}$ (Fig. 5a). The result showed that PPSR1 interacted with PSY1$^{130–412}$ and PSY1$^{130–238}$, but not PSY1$^{239–412}$ (Fig. 5b), indicating that the 130–238 amino acids in the PSY1 sequence are required for the interactions between PSY1 and PPSR1. It should be noted that, while PPSR1 interacts with N-terminal end of PSY1, the ubiquitination sites of PSY1 occur at the C-terminus of the protein. This could be explained by the characteristics of E3 ligases, which specifically recruit substrate proteins and transfer the activated ubiquitin from E2 enzymes to the substrates[14]. The E3 recognition regions and the ubiquitination sites in the substrates are likely to be different.

The LCI assay showed that the tobacco leaves co-expressing cLUC-PPSR1 and PSY1-nLUC generated luciferase activity, whereas the negative control exhibited no signal (Fig. 5c), confirming that PPSR1 interacts with PSY1. Semi-in vivo pull-down assay demonstrated that HA-tagged full-length PSY1 (PSY1-HA) can directly bind to MBP-tagged PPSR1 (MBP-PPSR1), but not MBP tag protein (Fig. 5d). To further verify the interactions between PPSR1 and PSY1, a Co-IP assay was carried out in tobacco leaves co-expressing Flag-PPSR1 and PSY1-HA. As shown in Fig. 5e, Flag-PPSR1 was immunoprecipitated with PSY1-HA by anti-HA agarose. Together, these data indicated that PPSR1 interacts with PSY1.

Fluorescence microscopy showed the green fluorescent signal from eGFP-tagged full-length PSY1 (PSY1-eGFP) merged the red fluorescent signal from mCherry-tagged PPSR1 (PPSR1-mCherry) in the cytosol (Fig. 5f), suggesting the subcellular colocalization of PSY1 and PPSR1. Since the mature PSY1 localized in the plastid, the fluorescent signal of PSY1-eGFP in the cytosol should be produced by the PSY1 precursor proteins.

We subsequently examined the ubiquitination of PSY1. The HA-tagged full-length PSY1 (PSY1-HA) was co-expressed with Flag-tagged ubiquitin (Flag-Ub) and PPSR1 in tobacco leaves, and then the total soluble proteins were extracted for ubiquitination assay. The formation of high molecular mass bands, which represent ubiquitinated PSY1-HA, was detected (Fig. 5g). This suggests that PPSR1 mediates the ubiquitination of PSY1. Notably, the ubiquitinated signal occurred over a band of the expected size (~48-kDa) for full-length PSY1 molecule (i.e., precursor) fused with HA tag. Due to the removal of the transit peptide (~7-kDa) from the PSY1 precursor during import into the plastid, the mature PSY1-HA protein, which could be detected using a robust protein extraction method (Supplementary Fig. 5), has a predicted molecular mass of ~41-kDa. Together, these data suggest that PPSR1 interacts with PSY1 precursor in the cytosol and mediates its ubiquitination.

**PPSR1-mediated degradation affects steady-state level of PSY1 protein**. The main function of protein ubiquitination is to mediate the degradation of substrate proteins through the 26S proteasome[27]. To investigate whether the ubiquitination sites of PSY1 led to its degradation, the HA-tagged full-length PSY1 (PSY1-HA) was co-expressed with Flag-tagged PPSR1 (Flag-PPSR1) in *N. benthamiana* leaves. As shown in Fig. 6a, the intensity of PSY1-HA was substantially decreased in the presence of Flag-PPSR1. The decrease in PSY1-HA could be rescued when the proteasome inhibitor MG132 was applied. These results indicated that PPSR1 mediates PSY1 degradation via the ubiquitin–proteasome system.

We then set out to determine PSY1-HA stability by monitoring the degradation rate of PSY1-HA in the presence of translation inhibitor cycloheximide (CHX). As shown in Fig. 6b, c, PSY1-HA degraded quickly in *N. benthamiana* after CHX treatment. When PSY1-HA was co-expressed with Flag-PPSR1, the degradation rate of PSY1-HA was increased (Fig. 6b, c), confirming that PPSR1 participates in the degradation of PSY1. Accordingly, when the identified ubiquitinated lysine residues (Lys380 and Lys406) were separately replaced by arginine (R) (Fig. 6d), the degradation rate of PSY1-HA was decreased (Fig. 6e, f). Mutation of both ubiquitinated residues further decreased the degradation rate of PSY1, suggesting that both Lys380 and Lys406 are essential for the ubiquitin-mediated protein degradation (Fig. 6e, f).

We next examined the protein levels of PSY1 in tomato fruit of *ppsr1* mutants and wild type. Proteins were extracted from the early stage of fruit ripening (34 DPA) to exclude the interference of fruit developmental state. More PSY1 proteins were accumulated in the fruit of *ppsr1* mutants, compared with the fruit of wild type (Fig. 6g). Quantification of PSY1 protein levels showed ~3-fold increases in the *ppsr1* mutants compared with the wild-type control (Fig. 6g). *PSY1* transcript levels at the same ripening stage were also measured using quantitative real-time PCR. No significant differences were observed between the wild-type and the *ppsr1* mutants (Fig. 6h). These data indicated that PPSR1-mediated protein degradation affected steady-state level of PSY1 protein, and the observed increase of PSY1 protein levels in the *ppsr1* mutants occurred post-translationally, but not a consequence of enhanced gene expression. Based on our results and previous studies, we propose a model for the regulation of carotenoid biosynthesis by PPSR1-mediated PSY1 precursor ubiquitination and degradation in ripening tomato fruit (Fig. 6i).

## Discussion
**PPSR1-mediated protein ubiquitination modulates carotenoid biosynthesis**. Ubiquitin-mediated protein degradation participates in the regulation of various physiological processes in plants, such as development, hormone signaling, stress response, and disease resistance[27–31]. In this system, the E1–E2–E3

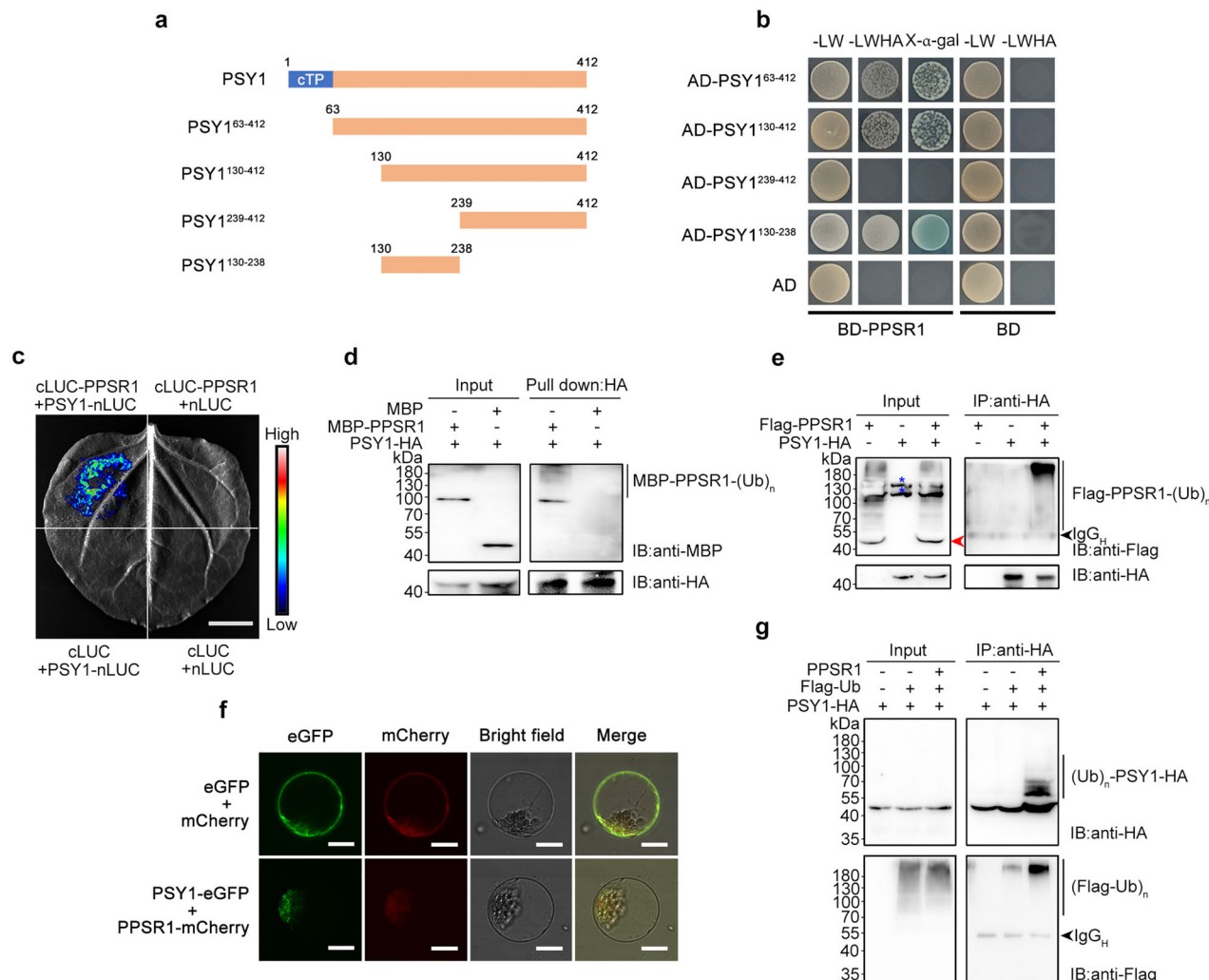

**Fig. 5 PPSR1 binds to PSY1 and mediates its ubiquitination. a, b** Y2H assay revealing the region of PSY1 that interacts with PPSR1. **a** Schematic illustration for full-length PSY1 protein and the truncated forms used in Y2H analysis. Numbers indicate the positions of the first and last amino acid in the sequences. cTP, chloroplast transit peptide. **b** The PPSR1 fused with the binding domain (BD) of GAL4 (BD-PPSR1) and the truncated PSY1 fused with the activation domain (AD) of GAL4 were co-expressed in yeast. The transformants were selected on SD/-Leu/-Trp (-LW) and SD/-Leu/-Trp/-His/-Ade (-LWHA) with or without X-α-gal. **c** LCI assay revealing the interactions between PSY1 and PPSR1. The PPSR1 fused with the C-terminus of LUC (cLUC-PPSR1) was co-expressed with the PSY1 fused with the N-terminus of LUC (PSY1-nLUC) in tobacco (*Nicotiana benthamiana*) leaves. Scale bar, 1 cm. **d** Semi-in vivo pull-down assay revealing the interactions between PPSR1 and PSY1. The recombinant MBP-PPSR1 and MBP (negative control) were mixed with PSY1-HA expressed in tobacco leaves, and incubated with anti-HA agarose. The eluted proteins were detected by immunoblot using anti-MBP and anti-HA antibodies, respectively. IB, immunoblot. **e** Co-IP assay revealing the interactions between PSY1 and PPSR1. The Flag-PPSR1 and PSY1-HA fusion proteins were co-expressed in tobacco leaves. **f** Subcellular colocalization of PSY1 and PPSR1. The PSY1-eGFP and PPSR1-mCherry fusion proteins were transiently co-expressed into tobacco leaves. The tobacco leaves expressing eGFP or mCherry were used as the negative control. Scale bars, 20 μm. **g** Ubiquitination assay of PSY1. The *Agrobacteria* carrying 35S::*PSY1-HA*, 35S::*Flag-ubiquitin* (*Ub*), and 35S::*PPSR1* constructs were infiltrated into the tobacco leaves. For (**e**) and (**g**), the total proteins were extracted from the infected leaves treated with MG132 and incubated with anti-HA agarose to enrich PSY1-HA. The eluted proteins were subjected to immunoblot using anti-Flag and anti-HA antibodies, respectively. The red arrowhead indicates the predicted Flag-PPSR1. The black arrowhead refers to heavy chain of antibody (IgG$_H$). Blue asterisks refer to nonspecific bands. IB, immunoblot; (Ub)$_n$, polyubiquitin chain.

ubiquitin enzymes cooperate to attach ubiquitin to the substrate proteins, which are then recognized and degraded by the 26S proteasome[27]. The 26S proteasome in plant cells is present in both the cytoplasm and the nucleus, and therefore ubiquitination was initially thought to occur only in proteins of these cellular compartments[27]. Later findings have indicated that ubiquitination and subsequent degradation also happens in proteins on cell surface membrane, or even in proteins from endoplasmic reticulum (ER) lumen and membrane[27]. By contrast, the regulation of ubiquitination on plastid proteins remains elusive. Intriguingly, recent researches unveiled that chloroplast outer membrane protein degradation was regulated by the ubiquitin–proteasome

system[11,32]. However, whether proteins in plastid metabolic pathways are mediated by ubiquitination remains unclear. In this study, we found that the E3 ubiquitin ligase PPSR1 targets precursor of PSY1, a key enzyme in the carotenoid biosynthesis pathway, and mediates its ubiquitination and degradation. Such proteolytic regulation changes the steady-state level of PSY1 protein, thereby modulating carotenoid biosynthesis. These data uncover a specific regulatory role of E3 ubiquitin ligase on plastid metabolic processes, which was achieved by modulating precursors of plastid-destined proteins in the biosynthetic pathways. This is inconsistent with previous observation showing that E3 ligase generally targets accumulated, unimported precursor

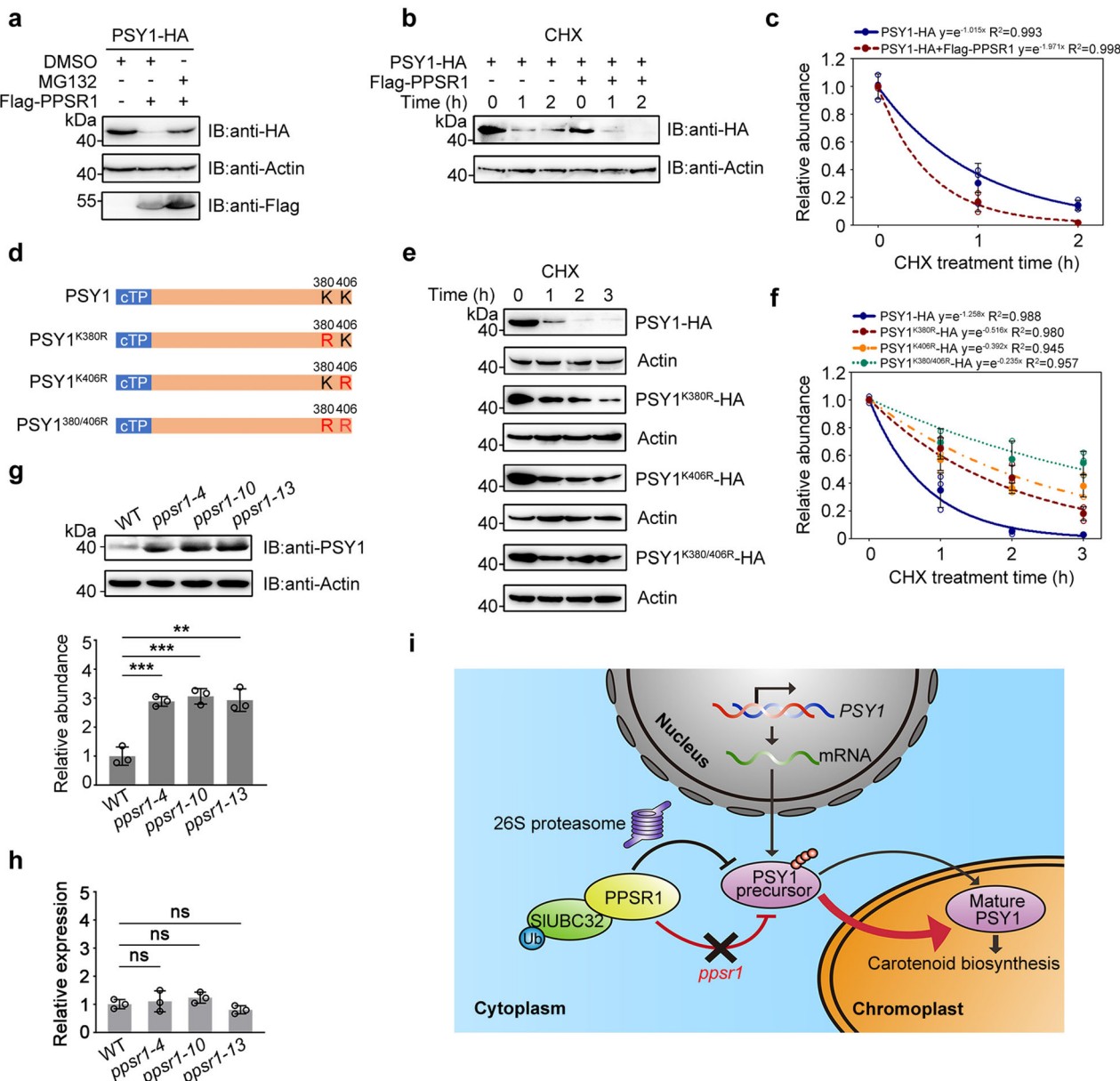

**Fig. 6 PPSR1 modulates PSY1 protein level via ubiquitination. a** Stability assay of PSY1. The PSY1-HA fusion protein was co-expressed with Flag-PPSR1 in tobacco (*Nicotiana benthamiana*) leaves. The total proteins were extracted and submitted to immunoblot using anti-HA and anti-Flag antibodies, respectively. The *N. benthamiana* actin was used as the loading control. IB, immunoblot. **b** Degradation rate assays of PSY1 under the action of PPSR1. The PSY1-HA and Flag-PPSR1 fusion proteins were co-expressed in tobacco leaves. The leaves were treated with translation inhibitor cycloheximide (CHX), and the total proteins were extracted for immunoblotting with anti-HA antibody at an indicated time point after treatment. **c** Quantification of protein levels in (**b**) by ImageJ. **d** Diagram showing PSY1 with ubiquitination site mutations. K, lysine; R, arginine. **e** Degradation rate assays of PSY1 and its mutated forms. The PSY1 and its mutated forms expressed in tobacco leaves were treated with CHX and submitted to immunoblot as described in (**b**). **f** Quantification of protein levels in (**e**) by ImageJ. **g** Expression of PSY1 protein in fruit of wild-type (WT) and *ppsr1* mutants. Total protein was extracted and submitted to immunoblot using anti-PSY1 antibody (upper panel). Equal loading was confirmed by an anti-actin antibody. The protein levels were quantified by ImageJ (lower panel). **h** Gene expression of *PSY1* in fruit of WT and *ppsr1* mutants. Total RNA was isolated and submitted to quantitative real-time PCR. The *ACTIN* gene was used as the internal control. For (**c**), (**f**), (**g**), and (**h**), error bars represent the means ± standard deviation (SD) of three independent experiments. The circles indicate individual data points. Asterisks indicate significant differences (**\*\*P* < 0.01, \*\*\**P* < 0.001; two-tailed Student's *t*-test). **i** The working model for PPSR1-mediated post-translational regulation of PSY1. PPSR1 directly interacts with SlUBC32 and mediates degradation of PSY1 precursor, which is nucleus-encoded and synthesized in the cytosol, via 26S proteasome. In the absence of PPSR1, more PSY1 precursor was transported into the plastid, leading to the accumulation of PSY1 protein. The increased PSY1 protein accelerates the biosynthesis of carotenoid in the plastid.

proteins[12]. Due to the importance of precursors on protein steady-state levels, we propose that E3 ligase-mediated protein ubiquitination and degradation may regulate various biological processes in plant cells by targeting the precursors of proteins involved in these biological processes.

The RING-type E3 ubiquitin ligases might mediate substrate degradation without the aid of chaperones, i.e., heat shock proteins (HSPs). We detected whether PPSR1 interacts with HSP70/HSP90, which have been reported to form a complex with Carboxy terminus of Hsc70-Interacting Protein (CHIP), an E3

ligase responsible for plastid-destined precursor degradation under stress conditions in *Arabidopsis*[12]. No interactions were observed between PPSR1 and HSP70/HSP90 (Supplementary Fig. 6). In addition, there were no apparent differences in protein levels of HSP70/HSP90 between fruit of wild-type and *ppsr1* mutants (Supplementary Fig. 7). These data suggest that PPSR1 mediates PSY1 precursor degradation in a HSP70/HSP90-independent manner. These characteristics of PPSR1 are similar to those of Misfolded Protein Sensing RING E3 ligase 1 (MPSR1), a RING-type E3 ligase that directly recognizes its substrates and mediates their degradation in the absence of chaperones or cofactors[20]. It should be noted that our study does not eliminate the possibility of other chaperones to interact with PPSR1.

**PSY1 is regulated at multiple levels**. As the main rate-determining enzyme, PSY acts in the first committed step in carotenogenesis and directs metabolic flux into the carotenoid biosynthetic pathway[33,34]. Constitutive overexpression of *PSY* has increased total carotenoid contents and substantially enhanced β-carotene synthesis in tissues of various crops, such as canola seeds, potato tubers, cassava roots, and tomato fruit[35–38]. Due to its central role in carotenoid biosynthesis, PSY is regulated at multiple levels. Substantial insights have been made into the transcriptional regulation of *PSY*. In *Arabidopsis* seedlings, the expression of the *PSY* gene is under the control of two transcription factors, PHYTOCHROME INTERACTING FACTOR 1 (PIF1) and LONG HYPOCOTYL 5 (HY5), which act antagonistically in photomorphogenesis[39,40]. *PSY* in tomato fruit is regulated by transcription factors of the MADS box family, such as RIPENING INHIBITOR (RIN) and FRUITFULL 1 (FUL1/TDR4), which bind directly to the promoter of *PSY* gene to induce its expression[41–44]. The expression of *PSY* gene in tomato is also reported to be feedback regulated by *cis*-carotenoids[45]. Recently, it has been revealed that PSY protein level is negatively controlled by carotenoid metabolites[46] and translational control elements located in the 5′ untranslated region (UTR) of *PSY* mRNAs[47]. Moreover, the proteostasis of PSY has been shown to be modulated by the cochaperone-like ORANGE (OR) protein and Clp protease[34,48]. However, it is not clear whether PSY is regulated by protein post-translational modification.

In the present study, we identified two ubiquitination sites (Lys380 and Lys406) in PSY1, the only PSY protein associated with carotenoid biosynthesis during tomato fruit ripening[49], and demonstrated that these ubiquitination sites are responsible for PSY1 precursor degradation. We speculate that PPSR1 modulates PSY1 protein ubiquitination and degradation mainly in unripe fruit to restrict the biosynthesis of carotenoids. At the onset of fruit ripening, the PPSR1-mediated degradation of PSY1 protein could be relieved due to uncharacterized reasons and the expression of *PSY1* gene is activated by various transcription factors. These findings provide insights into the regulation of carotenogenic enzymes and establish a link between protein ubiquitination and carotenoid biosynthesis.

**PPSR1 targets various proteins involved in fruit quality traits**. Besides PSY1, we identified a number of candidate substrates of PPSR1, such as E8, ADH2, alcohol acetyltransferase (AAT), and monooxygenase FAD-binding protein (FMO), which contain ubiquitination sites and changed abundance significantly in the *ppsr1* mutant (Supplementary Data 1 and 2). E8 is a homolog of 1-aminocyclopropane-1-carboxylate oxidase (ACO) essential for ethylene biosynthesis[50]. It is considered as a marker gene of tomato fruit ripening, since the transcript level of E8 increases dramatically during ripening[50]. Two ubiquitinated lysine residues were found in E8 (Supplementary Fig. 4), implying that the

protein level of E8 may be modulated by ubiquitination. Given that ethylene is involved in the generation of carotenoids[51], our data suggests that, in addition to directly targeting PSY1, PPSR1 might regulate carotenoid biosynthesis by mediating ubiquitination and degradation of E8, leading to the changes in ethylene accumulation. The functions of PPSR1 on proteolytic regulation of E8 and the relevant ethylene production deserve further investigation.

ADH2, AAT, and FMO are responsible for the formation of aroma volatiles in tomato fruit[52–55]. While ADH2 is related to the accumulation of hexanol and hexenol volatiles[52–54], AAT and FMO are associated with the formation of esters and phenylalanine-derived volatiles in ripe fruit, respectively[55]. Although aroma volatiles contribute greatly to the flavor quality, the regulation of aroma volatile biosynthesis in fruit remains largely unknown. Our data suggest that, by targeting components in the biosynthetic pathways, PPSR1 could be responsible for the generation of aroma volatiles in fruit. In conclusion, our results reveal a SlUBC32-PPSR1-PSY1 regulatory module which modulates carotenoid biosynthesis via the ubiquitin-mediated proteolytic pathway in tomato fruit. This regulatory mechanism may be applied to other metabolic pathways taken place in plastids, such as synthesis of fatty acids. Moreover, considering the multiple roles of protein ubiquitination and the similarity between plastids and mitochondria, which are also semi-autonomous organelles, we propose that the regulatory mechanism we describe here may exist in the regulation of metabolites that are synthesized in mitochondria.

## Methods

**Plant materials**. Seeds of wild-type tomato (*Solanum lycopersicum* cv. Ailsa Craig) were kindly provided by the Tomato Genetics Resource Center (TGRC, https://tgrc.ucdavis.edu/policy.aspx). The plants were cultivated in a greenhouse under standard culture conditions, which was supplied with regular fertilizer and supplementary lighting when required. To accurately follow fruit ages through development, flowers were tagged at anthesis. Wild-type fruit was harvested at mature green (MG), breaker (BR), orange (OR), and red ripe (RR), which were on average 34, 38, 41, and 45 days post-anthesis (DPA), respectively, based on the size, color, shape, and the development of seed and locular jelly in the fruit[56]. The mutants were collected at the equivalent ripening stages, as determined by the DPA. After harvesting, pericarps were immediately collected, frozen in liquid nitrogen, and stored at −80 °C until use.

**Generation of transgenic tomato plants**. CRISPR/Cas9 mediated gene-editing was carried out as described by Ma et al.[57] with minor modifications. In brief, four specific sgRNAs that targeted *PPSR1* were designed by CRISPR-P (version 2.0, http://crispr.hzau.edu.cn/CRISPR2). The expression cassettes (Target_1-Target_2 and Target_3-Target_4) driven by AtU3b and AtU3d promoters, respectively, were amplified and cloned into the pYLCRISPR/Cas9Pubi-H binary vector using the Golden Gate ligation method. The resulting constructs were transformed into *Agrobacterium tumefaciens* strain GV3101[58], which were subsequently infiltrated into the wild-type tomato cultivar Ailsa Craig[59]. Mutation on transgenic lines was verified by sequencing genomic regions flanking the target sites. The potential off-targets were predicted by CRISPR-P. All primers used to generate these constructs are listed in Supplementary Data 3.

**RNA isolation and quantitative real-time PCR**. Total RNA was isolated from tomato according to the method described by Moore et al.[60]. All tissues were ground into powder and mixed with extraction buffer containing 100 mM Tris-HCl, pH 8.0, 100 mM LiCl, 10 mM EDTA, 1% SDS, and 50% water-saturated phenol. RNA was precipitated with 4 M LiCl and collected by centrifugation at 12,000 × g for 20 min at 4 °C. Genome DNA digestion and reverse transcription of the extracted RNA were performed using the PrimeScript$^{TM}$ RT Reagent Kit (Takara) according to the manufacturer's protocol. Quantitative real-time PCR (qRT-PCR) was carried out with SYBR® Premix Ex Taq$^{TM}$ (Tli RNaseH Plus) (Takara) using the StepOne Plus Real-Time PCR System (Applied Biosystems). PCR primers listed in Supplementary Data 3 were designed by QuantPrime (http://quantprime.mpimp-golm.mpg.de/). PCR amplification was performed in a volume of 20 μl with the following program: 95 °C for 10 min, followed by 40 cycles of 95 °C for 15 s and 60 °C for 30 s. The cycle threshold (Ct) $2^{(-\Delta Ct)}$ method was applied to the relative quantification of gene transcription levels[61]. *ACTIN* (Solyc11g005330) was used to normalize the expression values. Three independent biological replicates with three technical repeats each were conducted.

**Y2H analysis**. Y2H screening was performed as described by Wang et al.[62]. The tomato cDNA library constructed in the prey vector pGADT7 (AD) was screened with the *SlUBC32* cDNA fragment cloned into the bait vector pGBKT7 (BD) in *Saccharomyces cerevisiae* strain AH109 (Clontech). The yeast zygote was selected on SD/-Leu-Trp-His-Ade medium (-LWHA) supplemented with α-D-galactoside (X-α-gal) according to the manufacturer's instructions (Clontech). The positive clones carrying putative SlUBC32-interacting proteins were identified by sequencing.

Y2H analysis was carried out using Matchmaker GAL4 Two-Hybrid System 3 following the manufacturer's protocols (Clontech). The cDNA fragments of the proteins were cloned into the AD and BD vectors, respectively. The primers used for the vector construction are listed in Supplementary Data 3. The resulting constructs were co-transformed into *S. cerevisiae* strain, and then plated on SD/-Leu-Trp medium (-LW) and SD/-Leu-Trp-His-Ade medium (-LWHA) containing X-α-gal. The transformants carrying empty vectors (BD or AD) were used as negative controls.

**LCI assay**. LCI assay was carried out as described by Chen et al.[63]. The coding sequence of *SlUBC32*, *PPSR1*, and *PSY1* was amplified from tomato cDNA and separately ligated into the pCambia1300–cLUC/nLUC vectors, which were kindly provided by Prof. Jianmin Zhou (Institute of Genetics and Developmental Biology, Chinese Academy of Sciences). The resulting constructs were separately introduced into *A. tumefaciens* strain GV3101[58]. After incubation at 28°C for 24 h, the *Agrobacteria* were pelleted by centrifugation and resuspended in infiltration buffer (10 mM MES, pH 5.6, 10 mM MgCl₂, and 100 μM acetosyringone) to a final OD₆₀₀ of 1.0. Then, the *Agrobacteria* suspension containing recombinant cLUC constructs was mixed with equal volume of *Agrobacteria* suspension containing recombinant nLUC constructs and kept at room temperature for 3 h without shaking. The mixture of *Agrobacteria* was then infiltrated into *N. benthamiana* leaves[64]. After culture for 48 h, the leaves were sprayed with 1 mM luciferin dissolved in ddH₂O containing 0.01% Triton X-100 and kept in dark for 5 min. Images of luminescence were captured using a chemiluminescence imaging system (Tanon). The tobacco leaves transformed with empty vectors expressing cLUC or nLUC were considered as negative controls. The results were repeated in at least three tobacco leaves. The primers used for the generation of LCI constructs are listed in Supplementary Data 3.

**Preparation of recombinant proteins and in vitro pull-down assay**. To generate MBP-tagged PPSR1 (MBP-PPSR1) and mtPPSR1 (MBP-mtPPSR1) recombinant proteins, the coding sequence of *PPSR1* and *mtPPSR1* was amplified and individually cloned into the pETMALc-H vector[65]. The mtPPSR1 was prepared by site-directed mutagenesis using the QuikChange II XL site-directed mutagenesis kit (Agilent Technologies) following the manufacturer's instructions. To obtain HA-tagged SlUBC32 (SlUBC32-HA), the coding sequence of *SlUBC32-HA* was amplified from vector containing 35S::*SlUBC32-HA* as describe below and inserted into the pET-30a vector (Merck KGaA). The resulting plasmids were then transformed into *E. coli* strain BL21 (DE3) competent cells. The recombinant protein expression and purification were performed as described by Zhou et al.[66]. The concentration of the recombinant proteins was determined[67]. The primers used for site-directed mutagenesis and vector construction are listed in Supplementary Data 3.

To examine the interactions between SlUBC32 and PPSR1 in vitro, 1 μg of recombinant SlUBC32-HA purified from *E. coli* was mixed with 500 ng of recombinant MBP-PPSR1 or MBP (negative control). The mixture was then incubated with anti-HA agarose (Cell Signaling Technology) at 4 °C for 2 h in binding buffer containing 20 mM Tris-HCl, pH 8.0, 100 mM NaCl, 1 mM EDTA, 1 mM PMSF, and 1% Triton X-100. After collection by spin columns, the beads were washed three times with binding buffer. The proteins were eluted with 1× SDS loading buffer at 95 °C for 5 min, and then subjected to immunoblot using anti-MBP (Beijing Protein Innovation) and anti-HA (Abmart) antibodies, respectively, as below.

**Semi-in vivo pull-down assay**. Semi-in vivo pull-down assay was conducted as described by Hu et al.[68] with some modifications. The coding region of *PSY1* was amplified from tomato cDNA and cloned into the pCambia1300-MCS-HA vector to generate 35S::*PSY1-HA* construct. The resulting construct was transformed into *A. tumefaciens* strain GV3101[58], which was subsequently infiltrated into *N. benthamiana* leaves[64]. After infiltration for 48 h, the *N. benthamiana* leaves were collected. Total proteins were extracted from the leaves with 1 ml of extraction buffer containing 50 mM Tris-HCl, pH 7.5, 100 mM NaCl, 1 mM EDTA, 1% Triton X-100, 5% glycerol, 1 mM PMSF, 1× protease inhibitor cocktail, and 50 μM MG132. After centrifugation at 12,000 × g for 20 min at 4 °C, the supernatant containing HA-tagged PSY1 (PSY1-HA) was collected and mixed with 500 ng of recombinant MBP-PPSR1 or MBP (negative control). Then the mixture was incubated with anti-HA agarose (Cell Signaling Technology) at 4 °C for 2 h. The agarose beads were collected and washed three times with extraction buffer. The proteins eluted from the beads were then subjected to immunoblot with anti-MBP (Beijing Protein Innovation) and anti-HA (Abmart) antibodies, respectively, as below.

**Co-IP analysis**. Co-IP assay was carried out as described by Tang et al.[69] with some modifications. The coding region of *SlUBC32* and *PSY1* was amplified from tomato cDNA and cloned into the pCambia1300-MCS-HA vector, respectively, resulting in 35S::*SlUBC32-HA* and 35S::*PSY1-HA* constructs. The coding sequence of *PPSR1* was amplified and cloned into the pCambia1300-Flag-MCS vector to generate 35S::*Flag-PPSR1* construct. The resulting constructs were then separately introduced into *A. tumefaciens* strain GV3101[58]. The *A. tumefaciens*-mediated transient expression in *N. benthamiana* leaves was conducted as described above. After infiltration for 30 h, the tobacco leaves were treated with 50 μM MG132 for 6 h. Total proteins were extracted from the leaves with 1 ml of extraction buffer (50 mM Tris-HCl, pH 7.5, 100 mM NaCl, 1 mM EDTA, 1% Triton X-100, 5% glycerol, 1 mM PMSF, 1× protease inhibitor cocktail, and 50 μM MG132). After centrifugation at 12,000 × g for 20 min at 4 °C, the supernatant containing the proteins was immunoprecipitated with 20 μl of anti-HA agarose (Cell Signaling Technology) at 4 °C for 2 h. The agarose beads were collected and washed twice with extraction buffer. The proteins were eluted from the beads with 1× SDS loading buffer at 95 °C for 5 min, and then subjected to immunoblot using anti-HA (Abmart) and anti-Flag (MBL Life Science) antibodies, respectively, as below. The primers used for the generation of constructs are listed in Supplementary Data 3.

**Subcellular localization**. For colocalization analysis, the coding sequence of *SlUBC32* was amplified from tomato cDNA and inserted into the pCambia1300-MCS-mCherry vector to produce 35S::*SlUBC32-mCherry* plasmid. The coding sequence of *PPSR1* and *PSY1* was amplified and individually cloned into the pCambia1300-MCS-eGFP or pCambia1300-MCS-mCherry vector to generate 35S::*PPSR1-eGFP*, 35S::*PPSR1-mCherry*, and 35S::*PSY1-eGFP* constructs. The resulting plasmids were transformed into *A. tumefaciens* strain GV3101[58], which was subsequently infiltrated into *N. benthamiana* leaves[64]. The *N. benthamiana* plants co-expressing mCherry-tagged SlUBC32 (SlUBC32-mCherry) and eGFP-tagged PPSR1 (PPSR1-eGFP) or co-expressing mCherry-tagged PPSR1 (PPSR1-mCherry) and eGFP-tagged PSY1 (PSY1-eGFP) were generated. The plants were cultured in the greenhouse for 36 h, and then the mesophyll protoplasts were isolated[70] and observed using a Leica confocal microscope (Leica DMI600CS). The primers used for vector construction are listed in Supplementary Data 3.

**In vitro ubiquitination assay**. The in vitro ubiquitination assay was performed as described by Xie et al.[71]. Briefly, 500 ng of purified MBP-PPSR1 or MBP-mtPPSR1 recombinant protein were mixed with 100 ng of E1 (UBA1, M55604.1) from wheat, 200 ng of E2 (UBCh5b, U39317.1) from human or SlUBC32-HA, and 2 μg of ubiquitin (UBQ14, At4g02890) from *Arabidopsis* in 30 μl of reaction buffer (50 mM Tris-HCl, pH 7.5, 5 mM MgCl₂, 2 mM ATP, and 2 mM dithiothreitol) at 30 °C for 2 h. The reaction was stopped with 1× SDS loading buffer at 95 °C for 5 min. The reaction products were submitted to immunoblot using anti-ubiquitin (P4D1, Santa Cruz Biotechnology) and anti-MBP (Beijing Protein Innovation) antibodies, respectively, as below. The *E. coli* strains expressing E1, E2, and ubiquitin were kindly provided by Prof. Jingbo Jin (Institute of Botany, Chinese Academy of Sciences).

**Cell-free degradation assay**. The cell-free degradation assay was performed as described by Wang et al.[72] with some modifications. Total proteins were extracted from tomato leaves with extraction buffer containing 25 mM Tris-HCl, pH 7.5, 10 mM NaCl, 10 mM MgCl₂, 1 mM PMSF, 5 mM dithiothreitol, and 2 mM ATP. After centrifugation at 16,000 × g for 15 min at 4 °C, the supernatant was recovered, and the protein concentration was determined[61]. Then, the purified recombinant MBP-PPSR1 or MBP-mtPPSR1 proteins (500 ng) were added into 500 μl of the total protein extracts (1 μg μl⁻¹ total protein) containing 500 ng purified MBP as loading control. The mixtures were reacted at 30 °C for 0, 1, 1.5, 2, 2.5, 3, 3.5, and 4 h, and then subjected to immunoblot analysis using anti-MBP antibody (Beijing Protein Innovation), as below. The band intensity was quantified using ImageJ software (https://imagej.nih.gov/ij/index.html) as described by Girish and Vijayalakshmi[73].

**iTRAQ-based quantitative proteomic analysis**. Proteins were extracted from wild-type and *ppsr1* mutant fruit at 38 DPA using a phenol extraction method[74]. The isolated proteins were solubilized in lysis buffer (20 mM HEPES, pH 8.0, and 8 M urea), and the protein concentration was measured[67]. One hundred micrograms of proteins from each sample were reduced with 10 mM dithiothreitol, alkylated with 50 mM iodoacetamide, and digested with 10 ng μl⁻¹ trypsin overnight. The tryptic peptides were desalted on a Sep-Pak C18 column (Waters, Inc.), and then labeled with the iTRAQ Reagents 4-plex Kit (Applied Biosystems) according to the manufacturer's protocol. Two independent biological replicates were applied for the iTRAQ experiment. The iTRAQ-labeled samples were combined, lyophilized, and fractionated with high-pH reversed-phase chromatography as described by Wang et al.[62]. Forty-eight fractions were collected along with the separation and pooled into a total of six fractions. After drying and desalting, the peptides were analyzed by NanoLC–MS/MS using a NanoLC system (NanoLC–2D Ultra Plus, Eksigent) equipped with a Triple TOF 5600 Plus mass spectrometer (AB SCIEX)[16].

Protein identification and relative quantification were performed by the ProteinPilot™ 4.5 software (AB SCIEX). The mass spectra data were used to

search the *S. lycopersicum* protein database (ITAG2.4_proteins_full_desc.fasta). By using the Pro Group™ algorithm (AB SCIEX), the peptide for quantification was automatically selected to calculate the reporter peak area. To determine the global FDR for peptide identification, a reverse database search strategy was applied[75]. Only proteins identified below the 1% global FDR were utilized to calculate the meaningful cutoff value using a population statistics method applied to the biological replicates[76].

**Ubiquitinated peptide enrichment and identification**. Ubiquitinated peptide enrichment was performed using the PTMScan Ubiquitin Remnant Motif (K-ε-GG) kit according to the manufacturer's protocol (Cell Signaling Technology). Briefly, proteins were extracted from wild-type and *ppsr1* mutant fruit at 38 DPA and solubilized in lysis buffer as described above. Approximately 10 mg of the isolated proteins were reduced with 10 mM dithiothreitol, alkylated with 50 mM iodoacetamide, and digested with 10 ng μl$^{-1}$ trypsin overnight. After desalted on a Sep-Pak C18 column (Waters, Inc.), the tryptic peptides were lyophilized under vacuum and redissolved in immunoprecipitation buffer (IAP) containing 50 mM MOPS-NaOH buffer, pH 7.2, 10 mM Na$_2$HPO$_4$, and 50 mM NaCl, and then incubated with anti-K-(GG) antibody beads (Cell Signaling Technology) for 2 h at 4 °C. The beads were collected and washed twice with IAP buffer. The ubiquitinated peptides were eluted from the beads with 0.15% trifluoroacetic acid (TFA), desalted with C18 Stage Tips (Thermo Scientific), and analyzed by NanoLC–MS/MS.

Quantitative analysis of ubiquitinated peptides between wild-type and *ppsr1* mutant was performed using the SWATH-MS method[23]. Mass spectra were acquired on a TripleTOF 5600 Plus instrument (AB SCIEX) operating in the SWATH mode as described by Wang et al.[16]. The peptides and their ubiquitination sites were identified using ProteinPilot software (AB SCIEX) against the *S. lycopersicum* protein database (ITAG2.4_proteins_full_desc.fasta). The MS/MS spectra of the identified peptides were used to generate a library for SWATH processing and quantification by Peakview software (AB SCIEX). The library correlated both peptide identification and LC retention times to extract specific MS/MS transition data for each peptide. For each individual sample, the ion transitions from the ubiquitinated peptides were applied to retrieve quantitative data (in counts/s) using a 0.05 Da extraction width over a ± 5 min LC time and visualized with MarkerView (AB SCIEX). After normalizing using Total Area Sums, the extracted ions for selected peptides were analyzed using Student's *t*-test within Markerview (AB SCIEX) with three technical replicates for each sample. A *P* value < 0.05 was considered to be significant. The experiment was performed with two independent biological replicates.

**Preparation of polyclonal antibodies**. For PPSR1-specific antibody preparation, the coding region of *PPSR1* lacking the conserved domain was amplified from tomato cDNA and cloned into the pET-30a vector (Merck KGaA), which was then transformed into *E. coli* BL 21 (DE3). The recombinant protein expression was performed as described above. Then, the recombinant PPSR1 protein was purified by Ni-NTA His-Bind Resin according to the manufacturer's instructions (Merck KGaA), followed by further purification using 12% SDS-PAGE. The recombinant protein was excised from the gel and used to immunize rabbits at the Abmart Shanghai Co., Ltd (http://www.ab-mart.com.cn). The PPSR1-specific polyclonal antibody was affinity-purified from antisera by the AminoLink Plus Coupling Resin following the purification manual (Thermo Scientific). To raise polyclonal PSY1 antibody, a synthetic peptide KSLVPPTKTASL was used to inject rabbits, followed by affinity-purification using the synthetic peptide. The primers used for vector construction are listed in Supplementary Data 3. The specificity of the antibodies was confirmed by immunoblot assay (Supplementary Fig. 8).

**Immunoblot analysis**. For immunoblot analysis, proteins were extracted from tobacco leaves as described above or from tomato fruit using a phenol extraction method[74]. Protein samples were separated by 10% SDS-PAGE and then transferred to an Immobilon-P PVDF membrane (Millipore, IPVH00010) using a semi-dry transfer unit (Amersham, TE77). The membranes were blocked for 1 h at room temperature with 5% non-fat milk in TBST buffer. The immunoblotting was conducted with anti-HA, anti-Flag, anti-MBP, anti-actin, anti-ubiquitin, anti-PPSR1, or anti-PSY1 antibodies at room temperature for 1 h, followed by incubation with horseradish peroxidase (HRP)-conjugated anti-rabbit or mouse IgG secondary antibody (1:5000) at room temperature for another 1 h. The membranes were then washed four times with TBST buffer, and the immunoreactive bands were visualized by using chemiluminescence detection kit (SuperSignal®, Pierce Biotechnology). Unprocessed original blot images are provided in Supplementary Figs. 9 and 10.

**Carotenoid measurement**. Pericarp carotenoids were extracted and quantified as described by Xiong et al.[77] with some modifications. In brief, the pericarps of fruit (2 g) powdered in liquid nitrogen was added to 20 ml of extraction buffer (hexane: acetone:ethanol at 2:1:1 by volume) containing 0.1% butylated hydroxytoluene (BHT). The mixture was vortexed for 30 min until the sample was decolored, followed by centrifugation at 4000 × *g* for 10 min. The supernatant was then collected and dried under a stream of nitrogen gas. The dried residue was resuspended

in 2 ml of methyl tert-butyl ether (MTBE) containing 0.1% BHT. The resuspended sample was saponified with 10% KOH in methanol for 30 min and dried in a nitrogen stream. Then, the dried residue was redissolved with 1 ml MTBE containing 0.1% BHT and filtered through a 0.22 μm membrane to remove insoluble particles. All the procedures above were conducted in low light. The individual carotenoids (phytoene, lycopene, and β-carotene) were identified and quantified according to the retention time and dose–response curves of the HPLC grade standards (Supplementary Fig. 11) using ACQUITY UPC$^2$ System (Waters, Inc.) equipped with a C$_{18}$ column according to the manufacturer's recommendations. Each sample contained five fruits, and the experiment was performed with three independent biological replicates.

**In vivo ubiquitination assay**. For ubiquitination assay in vivo, the 35S::*PSY1-HA* construct was generated as described above. The coding region of ubiquitin was amplified and inserted into the pCambia1300-Flag-MCS vector to generate 35S::*Flag-ubiquitin* construct. The full-length coding sequence of *PPSR1* was ligated into the pCambia1300 vector to construct 35S::*PPSR1*. The resulting constructs were transformed into *A. tumefaciens* strain GV3101[58], and then expressed transiently in *N. benthamiana* leaves[64]. After 30 h of incubation, the agroinfiltrated leaves were treated with 50 μM MG132 for 6 h, and the total proteins were extracted from *N. benthamiana* leaves as described above. The extracts were centrifuged at 12,000 × *g* for 20 min at 4 °C, and then the supernatants were incubated with anti-HA agarose (Cell Signaling Technology) at 4 °C for 2 h. After washing three times with extraction buffer, the bound proteins were eluted with 1× SDS loading buffer at 95 °C for 5 min. The eluted proteins were subjected to immunoblot analysis using anti-Flag (MBL life science) and anti-HA (Abmart) antibodies, respectively. The primers used for vector construction are listed in Supplementary Data 3.

**Protein stability and degradation rate assays**. The protein stability assay was performed with a transient expression system in *N. benthamiana*. The 35S::*PSY1-HA* and 35S::*Flag-PPSR1* constructs were generated as described above. The resulting constructs were individually introduced into *A. tumefaciens* strain GV3101[58]. After cultivation, the Agrobacteria harboring the indicated constructs were infiltrated into *N. benthamiana* leaves[64]. Forty-two hours after infiltration, the leaves were treated with 50 μM MG132 or DMSO (negative control) for 6 h. The total proteins were then extracted from the *N. benthamiana* leaves as described above and subjected to immunoblot using anti-HA (Abmart), and anti-Flag (MBL Life Science) antibodies, respectively. Equal loading was confirmed with an anti-actin antibody (Abmart).

For degradation rate analysis, the agroinfiltrated *N. benthamiana* leaves were treated with 250 μM cycloheximide (CHX) after 36 h of incubation. The mutated form of PSY1 (PSY1$^{K380R}$, PSY1$^{K406R}$, and PSY1$^{K380/406R}$) was generated by site-directed mutagenesis using the QuikChange II XL site-directed mutagenesis kit (Agilent Technologies) following the manufacturer's instructions. The band intensity was quantified using ImageJ software (https://imagej.nih.gov/ij/index.html). Values represent the average of three independent replicates. The primers used for vector construction are listed in Supplementary Data 3.

**Statistics and reproducibility**. Statistical analyses of data were performed using Microsoft Excel and GraphPad Prism 8.0 software. The band intensity of western blot was quantified by ImageJ software. Results are shown as the means ± standard deviation (SD) of three independent biological experiments. Statistical significance was analyzed by two-tailed Student's *t*-test. *P*-values of 0.05 or less were considered statistical significance (*$P < 0.05$, **$P < 0.01$, ***$P < 0.001$) and shown in figures. Regression analyses were conducted by linear or exponential regression model. Source data underlying the graphs are provided in Supplementary Data 4.

**Reporting summary**. Further information on research design is available in the Nature Research Reporting Summary linked to this article.

## Data availability

All data that support the findings of this study are available from the corresponding author upon request. The mass spectrometry proteomics data for iTRAQ and ubiquitinated peptide identification have been deposited in the PRIDE archive (Nos. PXD018731 and PXD018707, respectively; https://www.ebi.ac.uk/pride/archive).

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

## Acknowledgements

We would like to thank Z. Lu for analysis of MS/MS and J. Li for assistance with confocal microscopy. We thank J. Zhou (Institute of Genetics and Developmental Biology, Chinese Academy of Sciences) for providing the pCambia1300–cLUC/nLUC vectors, Y. Liu (South China Agriculture University) for providing the pYLCRISPR/Cas9Pubi-H binary vector, and J. Jin (Institute of Botany, Chinese Academy of Sciences) for providing *E. coli* strains expressing E1, E2, and ubiquitin. We also thank the PRIDE team for the deposition of our mass spectrometry proteomics data to the ProteomeXchange Consortium. This work was supported by the National Natural Science Foundation of China (grant Nos. 31925035, 31930086, and 31572174).

## Author contributions

G.Q. designed the research. P.W., Y.W., and W.W. performed the experiments. T.C. and S.T. provided discussions. G.Q., P.W., and Y.W. analyzed the data. G.Q. and P.W. wrote the manuscript.

## Competing interests

The authors declare no competing interests.
