## [Peer Review File · Communications Biology]

Reviewers' comments:

Reviewer #1 (Remarks to the Author):

This interesting manuscript reports on the role of a novel E3 ubiquitin ligase during tomato fruit ripening. The presented data are extensive and of high quality, and they point to a role for the E3 (PPSR1) in the proteolytic regulation of the cytosolic precursor form of the plastidic carotenoid biosynthetic enzyme, PSY1. The function of the tomato E3 has not previously been reported, and even the function of its Arabidopsis homologue is unknown. While the action of the UPS on plastid precursor proteins was previously described in some detail, the relevant CHIP E3 ligase seems to target accumulated, unimported preproteins generally; in this manuscript, the authors argue that PPSR1 plays a more specific regulatory role. The conclusion that PPSR1 targets PSY1 is based on robust tests, including interaction studies, *in vitro* ubiquitination assays, and protein turnover assays, and the results look convincing. I have some specific comments as follows:

1. In Figure S1, it is shown that the mRNA level of PPSR1 is reduced gradually as the fruit ripens, which matches well with the proposed function of PPSR1. However, the protein level of PPSR1 is very stable over the same timeframe, which is puzzling. This needs to be carefully explained.
2. E8 is another candidate target of PPSR1, based on the iTRAQ and ubiquitinome data. Thus, I am wondering why the authors did not investigate the role of PPSR1 in regulating E8. The E8 protein is a key factor regulating fruit ripening through ethylene biosynthesis. Therefore, the mRNA levels of PSY1 need to be checked in the *ppsr1* mutant lines, to determine their contribution to the levels of the PSY1 protein. This is important, as it has a bearing on the central conclusion and model of this study.
3. The main evidence that PPSR1 acts on the precursor form of PSY1 and not the mature form seems to be the cytosolic localization of PPSR1. I agree that this is significant. Nonetheless, in Figure 5, I found the colocalization and interaction analyses using an artificial mature-PSY1 construct to be rather contrived (5c-g). And, I was puzzled by the fact that there was no interaction with the precursor of PSY1 in the Y2H experiment (5b). This point needs better explanation and support.
4. The authors present evidence that the PSY1 ubiquitination sites are at the C-terminus of the protein; but show that PPSR1 only interacts with the N-terminal end of PSY1 (by Y2H). I can see how the interaction and modification sites could be different, but this point should be carefully discussed.
5. The Discussion section is rather thin. The iTRAQ quantitative proteomics and ubiquitinome assays both identified several other proteins influenced by PPSR1, which should be discussed thoroughly. Also, the biological significance of PPSR1 should be discussed; for example, why does PSY1 need to be regulated by protein degradation, rather than by transcriptional control?

Reviewer #2 (Remarks to the Author):

In this manuscript, Wang et al reported a novel post-translational regulatory mechanism for controlling carotenoid biosynthesis. Firstly, the authors identified the E3 PPSR1 as the regulator of carotenoid biosynthesis, loss-function of which via gene editing resulted in an increase accumulation

of carotenoids in tomato fruit. Then authors found the 18 ubiquitination targets of PPSR1 by a K-(GG) peptide immunoprecipitation coupling with quantitative proteomic analysis. PSY1, the key rate-limiting enzyme in the carotenoid biosynthetic pathway, which precursor is the target of PPSR1. Furthermore, Authors fully demonstrated that PPSR1-mediated degradation affected steady-state level of PSY1 protein in tomato fruit. These results significantly expand our knowledge about the multi-level regulation on carotenoid biosynthesis. Before publishing in CB, I suggest followings;

- 1.The subcellular of PPSR1 is in the cytosol and nucleus. SO are there some nucleus targets of PPSR1 in K-(GG) peptide immunoprecipitation?
- 2.In reference list, the number of authors for each reference is not consistent.
- 3.The figure legends are a little longer. It will better to put some information in the method section, not in the legends.
- 4.In figure 1, it is better that the panel a, b, c are in the same horizontal level.
- 5.In figure 1d and figure 5f, please give identification of the black arrowheads.
- 6.In figure 3a, the next WT sequence and the last chromatogram peak is too close.

Reviewer #3 (Remarks to the Author):

Overall, the paper was clearly structured and relatable in its experimental approach. Is was easy to follow and coherently explained. In the following there are some suggestions/critical comments: The abstract states that PPSR1 is responsible for carotenoid biosynthesis (page 2, lines 4-5). This phrasing is a little strong. Since you describe a situation, where PPSR1 regulates indirectly the amount of carotenoids, I would suggest to change the sentence into "PPSR1 is responsible for the regulation/affects the regulation of carotenoid biosynthesis".

In the results you introduce the term transit peptide without further explanation. It would be good to at least explain in one sentence what a transit peptide is. This would fit in the experimental part or in the introduction, since you write about precursor proteins in the introduction. In general, the utilization of the term "precursor" is not very precise throughout the entire paper, which is noted in detail later. You should give at least a brief definition of your understanding of the term precursor.

The characterization of PPSR1 was comprehensible and thorough regarding its interaction with the SIUB32, their colocalization and the self-ubiquitination activity and dimerization of PPSR1. The text (page 5, lines 24-27) creates the impression that a self-ubiquitination is a requirement to characterize a protein as E3-Ligase, I would like to know if that is true? Does every E3 ligase have to have self-ubiquitination activity? In figure 1d you do not mention the obviously very different amounts of Flag-PPSR1 in the inputs of your Co-IP. It would be good to discuss this and to give a first hint, that the PPSR1 might underlay (self-) ubiquitination.

The self-ubiquitination assays in figure 2b and 2c do not show a loading control of the E1/E2 and SIUB32, respectively. This would be good to exclude the possibility, that the results are resulting from different amounts of E1/E2 in the probes. Since E1 and E2 are overexpressed and purified from E. coli they surely have a detectable tag and in the methods you describe to have used SIUB32-HA. Figure 2g does not mention the number of replicates that were done to produce the diagram.

In figure 3 you introduce the ppsr1-mutants and show the very impressive effect of the PPSR1 knockout on the fruit ripening. For figure 3b it would be interesting to show the whole anti-PPSR1

Blot, because the first association would be that there exist truncated versions of the protein in the total protein extract. What molecular weight would be expected for these truncated constructs? The text (page 7, lines 11-12) mentions that all mutants were predicted to cause premature termination of PPSR1 protein translation within the following 40 bp sequence of editing sites. Supplementary figure 8 shows some unspecific bands, especially for total protein extracts. Would truncated PPSR1 be even distinguishable?

The phrase on page 10, lines 1-2 (“These results [...] in the plastids.”) could be optimized regarding the terminology of mPSY1 and mature PSY1. There should be a clear differentiation of the mPSY1 (I suppose “m” stands for mature, this should also be edited in the text and/or figures) and the kind of mature protein that occurs, when precursor protein PSY1 is proteolytically truncated after the import into the chloroplast.

In figure 5b you show clearly that BD-PPSR1 interacts with AD-mPSY1, but not with AD-PSY1. This is very puzzling to me and is not mentioned in the text. What is your explanation for this result? Is this what you had expected? It would be interesting to show one different interaction assay, such as the ones you did before (Pull-down, Co-IP...) to see, if the results remain the same or if the Y2H approach simply is not suitable for some reasons. Based on this assay you write in page 10, lines 4-5: “We used mPSY1 for subsequent analyses to mimic PSY1 precursor in the cytosol.” Here I am referring to my comment from the beginning, regarding the definition of a precursor. Of course, you describe that you mimic PSY1 precursor, but it must be noted that mPSY1 is not at all a precursor. Mature PSY1, lacking its transit peptide, should not be present in the cytosol. What about the folding situation of the protein in comparison to the precursor? How would you justify using the mature protein instead of the real precursor, that would be present in the cytosol in plants? This must be addressed! Is the one Y2H assay the only reason you decided to continue your following experiments with mPSY1 and not PSY1? And what is explanation for the results in the supplementary figures 4 and 5? I strongly recommend to do further experiments on the interaction between PPSR1 and full length PSY1 as suggested above.

In figure 5g you examined the ubiquitination of mPSY1-HA after co-expression with Flag-UB. You state that mPSY1-HA showed increased ubiquitination levels in the presence of PPSR1. It would be good to show the complete anti-HA immunoblot in the lower lane. The question that immediately arose was: Is the anti-Flag immunoblot sufficient to prove increased levels of ubiquitinated mPSY1-HA? Are the mPSY1 signals not mixed with self-ubiquitinated PPSR1? How do you differentiate that and should the signal pattern for the anti-HA blot not be similar to the anti-Flag blot, like it is shown in the self-ubiquitination assays? Or is the mPSY1 so much diluted by the polyubiquitination that it is not detectable?

I would find it reasonable to perform the experiments from figure 6 with PSY1 instead of mPSY1. In figure 6d you did not mention the number of replicates you used for the diagram.

On page 10, lines 23-25 is written: “To investigate whether PPSR1-mediated ubiquitination of PSY1 precursor led to its degradation, the HA-tagged mPSY1 was co-expressed...”. I find this phrase to be contradictory because you just did not work with actual precursor proteins. The definition of a precursor and the differentiation in the text is too vague. You could simply use a different phrasing, like “to investigate whether the ubiquitination sites of PSY1 led to its degradation...” or something similar.

In conclusion, the manuscript gives an interesting insight on the regulation of carotenoid synthesis by ubiquitination of PSY1 through the PPSR1 E3-ligase and gives an insight on a possible regulatory

module that can potentially be applied for many other situations in the plant cell. It also depicts the importance of precursor modulation for the situation in plant cells, which could be also better addressed in the discussion.

We would like to thank the reviewers for their constructive and positive feedback. Based on the reviewers' suggestions we have carried out additional experiments and substantially revised the manuscript. We believe that our manuscript has been improved by incorporating and implementing the reviewers' comments. All changes made during the revision are indicated in the manuscript by using Track Changes.

Response to the comments of Reviewer #1:

This interesting manuscript reports on the role of a novel E3 ubiquitin ligase during tomato fruit ripening. The presented data are extensive and of high quality, and they point to a role for the E3 (PPSR1) in the proteolytic regulation of the cytosolic precursor form of the plastidic carotenoid biosynthetic enzyme, PSY1. The function of the tomato E3 has not previously been reported, and even the function of its *Arabidopsis* homologue is unknown. While the action of the UPS on plastid precursor proteins was previously described in some detail, the relevant CHIP E3 ligase seems to target accumulated, unimported preproteins generally; in this manuscript, the authors argue that PPSR1 plays a more specific regulatory role. The conclusion that PPSR1 targets PSY1 is based on robust tests, including interaction studies, *in vitro* ubiquitination assays, and protein turnover assays, and the results look convincing. I have some specific comments as follows:

Question 1: In Figure S1, it is shown that the mRNA level of PPSR1 is reduced gradually as the fruit ripens, which matches well with the proposed function of PPSR1. However, the protein level of PPSR1 is very stable over the same timeframe, which is puzzling. This needs to be carefully explained.

Response: We thank the reviewer for pointing this out. The stable state of PPSR1 protein level during fruit ripening may be due to its self-ubiquitination activity as shown in Figure 2b. At the early stage of fruit ripening, the mRNA level of *PPSR1* is high, but part of the translated PPSR1 protein undergoes degradation via self-ubiquitination due to the absence of substrate proteins. At the later stage of fruit ripening, although the mRNA level of *PPSR1* is decreased, the accumulation of substrate proteins such as PSY1 alleviates the self-ubiquitination of PPSR1 protein, thus maintaining the stable state of PPSR1. Similar results were observed for MPSR1, a RING-type E3 ligase in *Arabidopsis*, which is also more stable in the presence of its substrate (Kim et al., 2017. *Proc. Natl. Acad. Sci. USA*, 114: E10009-E10017). The explanation has been added on Page 6, line 24-30.

Question 2: E8 is another candidate target of PPSR1, based on the iTRAQ and ubiquitinome data. Thus, I am wondering why the authors did not investigate the role of PPSR1 in regulating E8. The E8 protein is a key factor regulating fruit ripening through ethylene

biosynthesis. Therefore, the mRNA levels of PSY1 need to be checked in the *ppsr1* mutant lines, to determine their contribution to the levels of the PSY1 protein. This is important, as it has a bearing on the central conclusion and model of this study.

Response: In this study, we found that loss of PPSR1 function leads to an increase in carotenoids in tomato fruit. A number of proteins, including PSY1 and E8, were identified as the candidate targets of PPSR1. We focused on PSY1 because it represents the key rate-limiting enzyme that directly determines carotenoid biosynthesis. The role of PPSR1 in regulating E8 would be investigated in our future work.

We have carried out additional experiments to compare the mRNA levels of *PSY1* between the wild-type and the *ppsr1* mutant lines. Total RNAs were extracted from the early stage of fruit ripening (34 DPA) to exclude the interference of fruit developmental state on gene expression. As shown in Figure 6e and f, there were no significant differences in *PSY1* mRNA levels between wild-type and the *ppsr1* mutants. By contrast, more PSY1 proteins were accumulated in the fruit of *ppsr1* mutants, compared with the fruit of wild-type. These data indicate that the observed increase of PSY1 protein levels in the *ppsr1* mutants occurred post-translationally, but not a consequence of enhanced gene expression. The results have been described on page 12, line 29 to page 13, line 9.

Question 3: The main evidence that PPSR1 acts on the precursor form of PSY1 and not the mature form seems to be the cytosolic localization of PPSR1. I agree that this is significant. Nonetheless, in Figure 5, I found the colocalization and interaction analyses using an artificial mature-PSY1 construct to be rather contrived (5c-g). And, I was puzzled by the fact that there was no interaction with the precursor of PSY1 in the Y2H experiment (5b). This point needs better explanation and support.

Response: We thank the reviewer for the comments. We have carried out another set of experiments using the full-length PSY1 instead of the artificial mature-PSY1 for the interaction assays (LCI assay and Co-IP assay) and colocalization analyses. The results indicated that PPSR1 interacts with full-length PSY1 and the subcellular colocalization occurs in the cytosol (Figure 5). The corresponding contents in the Results section have been revised (page 11, line 6-19).

For the Y2H experiment, we have realized that the negative result may be caused by the transit peptide of PSY1. The Y2H system is based on the yeast transcription factor GAL4, which needs to be transported into the nucleus of yeast cells via the nuclear localization sequence (NLS) to activate the downstream reporter genes. The transit peptide in PSY1 may interfere with the transport of the reassembled GAL4 into the nucleus, leading to the failure of the interaction assay. For plastid-localized proteins, transit peptide at its N-terminus was always removed in the Y2H assay as previously reported (Cui et al., 2012. *Plant Physiol.*, **158**: 693-707; Mao et al., 2015. *Proc. Natl. Acad. Sci. USA*, **112**: 4152-4157). Therefore,

full-length PSY1 with transit peptide was not suitable for Y2H analysis. Figure 5b has been revised by removing the result of full-length PSY1 with transit peptide. Furthermore, to verify the interactions between PSY1 and PPSR1, we have carried out additional experiment (semi-*in vivo* pull-down assay). The result showed that PSY1-HA can directly bind to MBP-PPSR1, but not MBP tag protein (Figure 5d). The following sentences have been added on page 11, line 8-10 to describe the result.

“Semi-*in vivo* pull-down assay demonstrated that HA-tagged full-length PSY1 (PSY1-HA) can directly bind to MBP-tagged PPSR1 (MBP-PPSR1), but not MBP tag protein (Fig. 5d).”

Question 4: The authors present evidence that the PSY1 ubiquitination sites are at the C-terminus of the protein; but show that PPSR1 only interacts with the N-terminal end of PSY1 (by Y2H). I can see how the interaction and modification sites could be different, but this point should be carefully discussed.

Response: According to your suggestion. The following sentences have been added on page 10, line 21-26 to discuss the difference between the interaction parts and the ubiquitination sites.

“It should be noted that, while PPSR1 interacts with N-terminal end of PSY1, the ubiquitination sites of PSY1 occur at the C-terminus of the protein. This could be explained by the characteristics of E3 ligases, which specifically recruit substrate proteins and transfer the activated ubiquitin from E2 enzymes to the substrates¹⁴. The E3 recognition regions and the ubiquitination sites in the substrates are likely to be different.”

Question 5: The Discussion section is rather thin. The iTRAQ quantitative proteomics and ubiquitinome assays both identified several other proteins influenced by PPSR1, which should be discussed thoroughly. Also, the biological significance of PPSR1 should be discussed; for example, why does PSY1 need to be regulated by protein degradation, rather than by transcriptional control?

Response: A paragraph has now been added in the Discussion section (page 15, line 28 to page 16, line 18) to discuss the function of PPSR1 on several other candidate targets, such as E8 and ADH2.

In addition, the following sentences have been added on page 15, line 20-24 to discuss the biological significance of PPSR1.

“We speculate that PPSR1 modulates PSY1 protein ubiquitination and degradation mainly in unripe fruit to restrict the biosynthesis of carotenoids. At the onset of fruit ripening, the PPSR1-mediated degradation of PSY1 protein could be relieved due to uncharacterized reasons and the expression of *PSY1* gene is activated by various transcription factors.”

Response to the comments of Reviewer #2:

In this manuscript, Wang et al reported a novel post-translational regulatory mechanism for controlling carotenoid biosynthesis. Firstly, the authors identified the E3 PPSR1 as the regulator of carotenoid biosynthesis, loss-function of which via gene editing resulted in an increase accumulation of carotenoids in tomato fruit. Then authors found the 18 ubiquitination targets of PPSR1 by a K-(GG) peptide immunoprecipitation coupling with quantitative proteomic analysis. PSY1, the key rate-limiting enzyme in the carotenoid biosynthetic pathway, which precursor is the target of PPSR1. Furthermore, Authors fully demonstrated that PPSR1-mediated degradation affected steady-state level of PSY1 protein in tomato fruit. These results significantly expand our knowledge about the multi-level regulation on carotenoid biosynthesis. Before publishing in CB, I suggest followings;

Question 1: The subcellular of PPSR1 is in the cytosol and nucleus. So are there some nucleus targets of PPSR1 in K-(GG) peptide immunoprecipitation?

Response: We have identified several nuclear proteins, including Histone H2B (Solyc05g055440), Zinc finger, C6HC-type (Solyc03g117860), and Zinc finger, UBP-type (Solyc07g008050) in the K-(GG) peptide immunoprecipitation. Their functions deserve further investigation.

Question 2: In reference list, the number of authors for each reference is not consistent.

Response: We thank the reviewer for pointing this out. The number of authors for each reference is chosen according to the reference format of the journal. All authors are included in the reference list unless there are more than five, in which case only the first author is given, followed by 'et al.'.

Question 3: The figure legends are a little longer. It will better to put some information in the method section, not in the legends.

Response: We have removed some information that has been described in the Method section from the legends of Figure 1, 2, 5, and 6.

Question 4: In figure 1, it is better that the panel a, b, c are in the same horizontal level.

Response: The panels a, b, and c in Figure 1 have been adjusted to the same horizontal level.

Question 5: In figure 1d and figure 5f, please give identification of the black arrowheads.

Response: "IgG_H" has been added to indicate the black arrowhead in Figure 1 and 5.

Question 6: In figure 3a, the next WT sequence and the last chromatogram peak is too close.

Response: Figure 3a has been modified.

Response to the comments of Reviewer #3:

Overall, the paper was clearly structured and relatable in its experimental approach. It was easy to follow and coherently explained. In the following there are some suggestions/critical comments:

Question 1: The abstract states that PPSR1 is responsible for carotenoid biosynthesis (page 2, lines 4-5). This phrasing is a little strong. Since you describe a situation, where PPSR1 regulates indirectly the amount of carotenoids, I would suggest to change the sentence into “PPSR1 is responsible for the regulation/affects the regulation of carotenoid biosynthesis”.

Response: According to your suggestion, the sentence (page 2, line 4-5) has been changed to “Here we show that a tomato E3 ubiquitin ligase, Plastid Protein Sensing RING E3 ligase 1 (PPSR1), is responsible for the regulation of carotenoid biosynthesis.”.

Question 2: In the results you introduce the term transit peptide without further explanation. It would be good to at least explain in one sentence what a transit peptide is. This would fit in the experimental part or in the introduction, since you write about precursor proteins in the introduction. In general, the utilization of the term “precursor” is not very precise throughout the entire paper, which is noted in detail later. You should give at least a brief definition of your understanding of the term precursor.

Response: The following sentence has been added in the Introduction (page 3, line 22-25) to explain what a transit peptide is.

“These plastid-targeted proteins are synthesized in the cytosol as preproteins (precursor proteins), which contain an N-terminal transit peptide that directs the precursor proteins into the plastid before being proteolytically removed^{11,12}.”

In addition, we have utilized full-length PSY1 (i.e. precursor) instead of mPSY1 for all the experiments. Figure 5 and 6 and the corresponding contents have been revised as described below.

Question 3: The characterization of PPSR1 was comprehensible and thorough regarding its interaction with the SIUB32, their colocalization and the self-ubiquitination activity and dimerization of PPSR1. The text (page 5, lines 24-27) creates the impression that a self-ubiquitination is a requirement to characterize a protein as E3-Ligase, I would like to know if that is true? Does every E3 ligase have to have self-ubiquitination activity? In figure 1d you do not mention the obviously very different amounts of Flag-PPSR1 in the inputs of

your Co-IP. It would be good to discuss this and to give a first hint, that the PPSR1 might underlay (self-) ubiquitination.

Response: Self-ubiquitination activity is not a requirement to characterize a protein as E3 ligase. To avoid misunderstanding, the text has been revised as follows (page 5, line 28 to page 6, line 7):

“To determine whether PPSR1 functions as an E3 ligase, the MBP-tagged recombinant PPSR1 protein (MBP-PPSR1) was purified from *E. coli* and subjected to *in vitro* ubiquitination assay by incubation with wheat E1, human E2, and *Arabidopsis* ubiquitin. The reaction products were detected by immunoblot analysis using anti-MBP and anti-ubiquitin antibodies, respectively. As shown in Fig. 2b and Supplementary Fig. 2, the signals of high molecular mass bands, which represent ubiquitinated proteins, were observed in the intact reaction system using both anti-MBP and anti-ubiquitin detection, but not in the absence of a single component. These data indicated that PPSR1 has E3 ubiquitin ligase activity *in vitro* and can catalyze its self-ubiquitination.”

To explain the obviously different amounts of Flag-PPSR1 in the inputs of Co-IP in Figure 1d, the following sentence has been added on page 5, line 9-10.

“Notably, co-expression resulted in an increase in amounts of Flag-PPSR1 in the input of Co-IP, which might be caused by the self-ubiquitination of PPSR1.”

Question 4: The self-ubiquitination assays in figure 2b and 2c do not show a loading control of the E1/E2 and SIUBC32, respectively. This would be good to exclude the possibility, that the results are resulting from different amounts of E1/E2 in the probes. Since E1 and E2 are overexpressed and purified from *E. coli* they surely have a detectable tag and in the methods you describe to have used SIUB32-HA. Figure 2g does not mention the number of replicates that were done to produce the diagram.

Response: According to your suggestion, we have detected the abundances of E1, E2, and SIUBC32 in the *in vitro* ubiquitination assays. The results indicated that equal amounts of E1/E2 and SIUBC32 were added in the assays, excluding the possibility that the results are resulting from different amounts of E1/E2 and SIUBC32 in the probes (Supplementary Figure 2).

Three biological replicates have been carried out to produce the diagram of Figure 2g and this has been indicated in the figure legends.

Question 5: In figure 3 you introduce the *ppsr1*-mutants and show the very impressive effect of the *PPSR1* knockout on the fruit ripening. For figure 3b it would be interesting to show the whole anti-PPSR1 Blot, because the first association would be that there exist truncated versions of the protein in the total protein extract. What molecular weight would be expected for these truncated constructs? The text (page 7, lines 11-12) mentions that all mutants were

predicted to cause premature termination of PPSR1 protein translation within the following 40 bp sequence of editing sites. Supplementary figure 8 shows some unspecific bands, especially for total protein extracts. Would truncated PPSR1 be even distinguishable?

Response: The whole anti-PPSR1 immunoblot has now been presented in Figure 3b, according to your suggestion. The predicted molecular weight of the full-length PPSR1 is ~38-kDa, and the truncated version of the PPSR1 protein is ~19-kDa (170 amino acids). A 38-kDa band corresponding to the predicted full-length PPSR1 was detected in the wild-type, whereas no bands were observed in the *ppsr1* mutants (Fig. 3b), indicating that the predicted truncated versions of PPSR1 did not generate and *PPSR1* was successfully knocked out in the mutants. This has been described in the revised manuscript (page 7, line 30 to page 8, line 4).

We have optimized the immunoblotting reaction conditions and the unspecific bands in Supplementary figure 7a have been successfully avoided.

Question 6: The phrase on page 10, lines 1-2 (“These results [...] in the plastids.”) could be optimized regarding the terminology of mPSY1 and mature PSY1. There should be a clear differentiation of the mPSY1 (I suppose “m” stands for mature, this should also be edited in the text and/or figures) and the kind of mature protein that occurs, when precursor protein PSY1 is proteolytically truncated after the import into the chloroplast.

Response: We have applied full-length PSY1 instead of mPSY1 for all the experiments. The phrase “mPSY1” has been deleted throughout the manuscript.

Question 7: In figure 5b you show clearly that BD-PPSR1 interacts with AD-mPSY1, but not with AD-PSY1. This is very puzzling to me and is not mentioned in the text. What is your explanation for this result? Is this what you had expected? It would be interesting to show one different interaction assay, such as the ones you did before (Pull-down, Co-IP...) to see, if the results remain the same or if the Y2H approach simply is not suitable for some reasons. Based on this assay you write in page 10, lines 4-5: “We used mPSY1 for subsequent analyses to mimic PSY1 precursor in the cytosol.” Here I am referring to my comment from the beginning, regarding the definition of a precursor. Of course, you describe that you mimic PSY1 precursor, but it must be noted that mPSY1 is not at all a precursor. Mature PSY1, lacking its transit peptide, should not be present in the cytosol. What about the folding situation of the protein in comparison to the precursor? How would you justify using the mature protein instead of the real precursor, that would be present in the cytosol in plants? This must be addressed! Is the one Y2H assay the only reason you decided to continue your following experiments with mPSY1 and not PSY1? And what is explanation for the results in the supplementary figures 4 and 5? I strongly recommend to do further experiments on the interaction between PPSR1 and full length PSY1 as suggested above.

Response: We thank the reviewer for the comments. We have realized that the negative

result in the Y2H experiment may be caused by the transit peptide of PSY1. The Y2H system is based on the yeast transcription factor GAL4, which needs to be transported into the nucleus of yeast cells via the nuclear localization sequence (NLS) to activate the downstream reporter genes. The transit peptide in PSY1 may interfere with the transport of the reassembled GAL4 into the nucleus, leading to the failure of the interaction assay. For plastid-localized proteins, transit peptide at its N-terminus was always removed in the Y2H assay as previously reported (Cui et al., 2012. *Plant Physiol.*, 158: 693-707; Mao et al., 2015. *Proc. Natl. Acad. Sci. USA*, 112: 4152-4157). Therefore, full-length PSY1 with transit peptide was not suitable for Y2H analysis. Figure 5b has been revised by removing the result of full-length PSY1 with transit peptide. Furthermore, to verify the interactions between PSY1 and PPSR1, we have carried out additional experiment (semi-*in vivo* pull-down assay). The result showed that PSY1-HA can directly bind to MBP-PPSR1, but not MBP tag protein (Figure 5d). The following sentences have been added on page 11, line 8-10 to describe the result.

“Semi-*in vivo* pull-down assay demonstrated that HA-tagged full-length PSY1 (PSY1-HA) can directly bind to MBP-tagged PPSR1 (MBP-PPSR1), but not MBP tag protein (Fig. 5d).”

For the interaction assays (LCI assay and Co-IP assay), we have carried out substantial additional experiments using the full-length PSY1 instead of the artificial mature-PSY1. The results indicated that PPSR1 interacts with full-length PSY1. Figure 5 has been revised and the results are now described on page 11, line 6-14.

Question 8: In figure 5g you examined the ubiquitination of mPSY1-HA after co-expression with Flag-UB. You state that mPSY1-HA showed increased ubiquitination levels in the presence of PPSR1. It would be good to show the complete anti-HA immunoblot in the lower lane. The question that immediately arose was: Is the anti-Flag immunoblot sufficient to prove increased levels of ubiquitinated mPSY1-HA? Are the mPSY1 signals not mixed with self-ubiquitinated PPSR1? How do you differentiate that and should the signal pattern for the anti-HA blot not be similar to the anti-Flag blot, like it is shown in the self-ubiquitination assays? Or is the mPSY1 so much diluted by the polyubiquitination that it is not detectable?

I would find it reasonable to perform the experiments from figure 6 with PSY1 instead of mPSY1. In figure 6d you did not mention the number of replicates you used for the diagram.

On page 10, lanes 23-25 is written: “To investigate whether PPSR1-mediated ubiquitination of PSY1 precursor led to its degradation, the HA-tagged mPSY1 was co-expressed...”. I find this phrase to be contradictory because you just did not work with actual precursor proteins. The definition of a precursor and the differentiation in the text is too vague. You could simply use a different phrasing, like “to investigate whether the ubiquitination sites of PSY1 led to its degradation...” or something similar.

Response: We have realized that the anti-Flag immunoblot signals may be interfered by the

self-ubiquitinated PPSR1. Figure 5g has been revised using full-length PSY1 instead of mPSY1, and the whole anti-HA immunoblot is presented. When PPSR1 was co-expressed with PSY1-HA, the signals of ubiquitinated PSY1-HA were successfully detected by the anti-HA antibody. The bands for ubiquitinated PSY1-HA occurred above the predicted molecular weight of full-length PSY1, thereby excluding the interference of the self-ubiquitination of PPSR1. Combined with the results of anti-Flag immunoblot, our data indicated that PPSR1 mediates the ubiquitination of PSY1. The corresponding contents in the Results section have been revised (page 11, line 20-25).

For Figure 6, we have carried out another set of experiments using full-length PSY1 instead of mPSY1. The results indicated that PPSR1 mediates PSY1 degradation via the ubiquitin-proteasome system. Figure 6 has been revised and the results are now described on page 12, line 7-28. Three biological replicates have been carried out to produce the diagram of Figure 6d and this has been indicated in the figure legends.

According to your suggestion, the sentence “To investigate whether PPSR1-mediated ubiquitination of PSY1 precursor led to its degradation, the HA-tagged mPSY1 (mPSY1-HA) was co-expressed with Flag-tagged PPSR1 (Flag-PPSR1) in *N. benthamiana* leaves.” has been changed to “To investigate whether the ubiquitination sites of PSY1 led to its degradation, the HA-tagged full-length PSY1 (PSY1-HA) was co-expressed with Flag-tagged PPSR1 (Flag-PPSR1) in *N. benthamiana* leaves.” (page 12, line 7-9).

Question 9: In conclusion, the manuscript gives an interesting insight on the regulation of carotenoid synthesis by ubiquitination of PSY1 through the PPSR1 E3-ligase and gives an insight on a possible regulatory module that can potentially be applied for many other situations in the plant cell. It also depicts the importance of precursor modulation for the situation in plant cells, which could be also better addressed in the discussion.

Response: Thank you for your positive comments. The following sentences have been added on page 14, line 5-12 to discuss the importance of precursor modulation for the function of a protein in plant cells.

“These data uncover a specific regulatory role of E3 ubiquitin ligase on plastid metabolic processes, which was achieved by modulating precursors of plastid-destined proteins in the biosynthetic pathways. This is inconsistent with previous observation showing that E3 ligase generally targets accumulated, unimported precursor proteins¹². Due to the importance of precursors on protein steady-state levels, we propose that E3 ligase-mediated protein ubiquitination and degradation may regulate various biological processes in plant cells by targeting the precursors of proteins involved in these biological processes.”

REVIEWERS' COMMENTS:

Reviewer #1 (Remarks to the Author):

The authors have addressed the comments raised previously, and performed some important new experiments. Thus, I think that this revised version is very much improved.

However, there are still some issues that need further clarification, as detailed below.

1. The authors now use prePSY1 instead of mature PSY1 for the analyses, which is good. However, I am quite puzzled by the fact that only the precursor form of PSY1-HA could be detected when this protein was transiently expressed in tobacco leaves. This issue needs to be explained. Is it possible that this is related to the analysis of a tomato protein in tobacco leaves? – given the relatedness of these species, this seems unlikely.

Related to this, on page 11, lines 25-29, the authors argue that it is the precursor form rather than the mature form that is ubiquitinated. However, this conclusion cannot be made if there is no mature form of PSY1-HA detected at all.

2. On page 5, line 9-10, I am not convinced by the argument that “co-expression resulted in an increase in amounts of Flag-PPSR1 in the input of Co-IP” due to “the self-ubiquitination of PPSR1.” Self-ubiquitination will indeed result in accumulation of high molecular weight signals (Fig. 1d), but it should not also increase the amount of the non-ubiquitinated form of PPSR1 (the red arrowhead presumably indicates the non-ubiquitinated form of PPSR1). In the experiment shown, the overall signal of FLAG-PPSR1 was higher when UBC32 was also expressed, which cannot be explained by self-ubiquitination.

Also, it is noteworthy that it was FLAG antibody, not ubiquitin antibody, that was used for the detection here; more definitive evidence of PPSR1 ubiquitination would require use of both.

3. On page 6, line 24-30, the new discussion text is good, but the reference provided in the rebuttal letter should be added.

Reviewer #2 (Remarks to the Author):

All my concerns have been addressed.

Reviewer #3 (Remarks to the Author):

This is the revised version of a manuscript I reviewed earlier. In my previous review I raised many critical points that were mostly concerned with data presentation and better explanations for the rationales for some experimental designs. The authors have addressed all of these concerns to a satisfactory extent.

We would like to thank the reviewers for their constructive and positive feedback. Based on the reviewers' suggestions we have revised the manuscript.

Comments of Reviewer #1:

Question 1: The authors now use prePSY1 instead of mature PSY1 for the analyses, which is good. However, I am quite puzzled by the fact that only the precursor form of PSY1-HA could be detected when this protein was transiently expressed in tobacco leaves. This issue needs to be explained. Is it possible that this is related to the analysis of a tomato protein in tobacco leaves? – given the relatedness of these species, this seems unlikely. Related to this, on page 11, lines 25-29, the authors argue that it is the precursor form rather than the mature form that is ubiquitinated. However, this conclusion cannot be made if there is no mature form of PSY1-HA detected at all.

Response: We thank the reviewer for pointing this out. In Fig. 5g, we aimed to test our hypothesis that the precursor form of PSY1 undergoes ubiquitination. Therefore, total soluble proteins, which contain PSY1 precursor protein, were extracted from tobacco leaves expressing PSY1 for the ubiquitination assay. The mature PSY1 protein was not detected, because it is located in the plastid and hard to be dissolved in the extraction buffer used for total soluble proteins. By using a robust protein extraction method such as phenol extraction method, the mature PSY1 protein could be detected (Supplementary Fig. 5). The following sentences have been revised to explain this issue.

Page 11, line 270-274: “When HA-tagged full-length PSY1 (PSY1-HA) was co-expressed with Flag-tagged ubiquitin (Flag-Ub) and PPSR1, the formation of high molecular mass bands, which represent ubiquitinated PSY1-HA, was detected (Fig. 5g).” has been changed to “The HA-tagged full-length PSY1 (PSY1-HA) was co-expressed with Flag-tagged ubiquitin (Flag-Ub) and PPSR1 in tobacco leaves, and then the total soluble proteins were extracted for ubiquitination assay. The formation of high molecular mass bands, which represent ubiquitinated PSY1-HA, was detected (Fig. 5g).”.

Page 11, line 276-279: “Due to the removal of the transit peptide (~7-kDa) from the PSY1 precursor during import into the plastid, the mature PSY1 protein fused with HA tag has a predicted molecular mass of ~41-kDa.” has been changed to “Due to the removal of the transit peptide (~7-kDa) from the PSY1 precursor during import into the plastid, the mature PSY1-HA protein, which could be detected using a robust protein extraction method (Supplementary Fig. 5), has a predicted molecular mass of ~41-kDa.”.

Question 2: On page 5, line 9-10, I am not convinced by the argument that “co-expression resulted in an increase in amounts of Flag-PPSR1 in the input of Co-IP” due to “the self-ubiquitination of PPSR1.” Self-ubiquitination will indeed result in accumulation of high

molecular weight signals (Fig. 1d), but it should not also increase the amount of the non-ubiquitinated form of PPSR1 (the red arrowhead presumably indicates the non-ubiquitinated form of PPSR1). In the experiment shown, the overall signal of FLAG-PPSR1 was higher when UBC32 was also expressed, which cannot be explained by self-ubiquitination. Also, it is noteworthy that it was FLAG antibody, not ubiquitin antibody, that was used for the detection here; more definitive evidence of PPSR1 ubiquitination would require use of both.

Response: We thank the reviewer for the comments. The following sentences (page 5, line 94-101) have been added to explain the increase in amount of the non-ubiquitinated form of PPSR1.

“Notably, high molecular weight signals over a band of the predicted Flag-PPSR1 (indicated by a red arrowhead) was observed in the input of Co-IP (Fig. 1d), which may be caused by the self-ubiquitination of PPSR1. Co-expression resulted in an increase in amounts of Flag-PPSR1 and this could be explained by the feedback regulation of PPSR1 by transcriptional regulators that are targeted by PPSR1. It is conceivable that co-expression decreases PPSR1 activity due to self-ubiquitination, thereby attenuating the degradation of the substrates including the transcriptional regulators, which in turn improve the expression of PPSR1.”

Question 3: On page 6, line 24-30, the new discussion text is good, but the reference provided in the rebuttal letter should be added.

Response: The reference provided in the rebuttal letter has been added on page 6, line 144.

Comments of Reviewer #2:

All my concerns have been addressed.

Response: We thank the Reviewer for the positive feedback.

Comments of Reviewer #3:

This is the revised version of a manuscript I reviewed earlier. In my previous review I raised many critical points that were mostly concerned with data presentation and better explanations for the rationales for some experimental designs. The authors have addressed all of these concerns to a satisfactory extent.

Response: We thank the Reviewer for the positive feedback.